# Clinal genomic analysis reveals strong reproductive isolation across a steep habitat transition in stickleback fish

Quiterie Haenel[1✉], Krista B. Oke [2,3], Telma G. Laurentino[1], Andrew P. Hendry[3] & Daniel Berner [1✉]

How ecological divergence causes strong reproductive isolation between populations in close geographic contact remains poorly understood at the genomic level. We here study this question in a stickleback fish population pair adapted to contiguous, ecologically different lake and stream habitats. Clinal whole-genome sequence data reveal numerous genome regions (nearly) fixed for alternative alleles over a distance of just a few hundred meters. This strong polygenic adaptive divergence must constitute a genome-wide barrier to gene flow because a steep cline in allele frequencies is observed across the entire genome, and because the cline center closely matches the habitat transition. Simulations confirm that such strong divergence can be maintained by polygenic selection despite high dispersal and small per-locus selection coefficients. Finally, comparing samples from near the habitat transition before and after an unusual ecological perturbation demonstrates the fragility of the balance between gene flow and selection. Overall, our study highlights the efficacy of divergent selection in maintaining reproductive isolation without physical isolation, and the analytical power of studying speciation at a fine eco-geographic and genomic scale.

[1] Department of Environmental Sciences, Zoology, University of Basel, Basel, Switzerland. [2] College of Fisheries and Ocean Sciences, University of Alaska Fairbanks, Juneau, AK, USA. [3] Redpath Museum and Department of Biology, McGill University, Montréal, QC, Canada. ✉email: quiterie.haenel@unibas.ch; daniel.berner@unibas.ch

Deciphering the origin of species requires understanding the nature of reproductive isolation between diverging populations[1–4]. At the genomic level, reproductive isolation is typically studied through the marker-based comparison of populations that diverged relatively recently into ecologically different habitats[5–8]. Genome regions showing exceptionally strong population differentiation are inferred to harbor loci important to adaptive divergence that potentially also restrict the exchange of genetic material across larger chromosome segments, or the genome as a whole[9]. This common analytical approach generally pays insufficient attention to the fine-scale eco-geography of speciation; our understanding of the genomics of reproductive isolation can benefit greatly from investigating diverging populations across their contact zones at a high geographic resolution[10–15]. One reason is that such a clinal focus can reveal over what distance gene flow between populations occurs. Moreover, if marker resolution is sufficiently high – ideally whole-genome resolution, we can learn to what extent gene flow differs among genome regions. These details are crucial for evaluating the strength and genetic architecture of reproductive isolation. Another benefit is that fine-grained clinal analyses facilitate recognizing a possible link between reproductive isolation and ecological transitions, and hence the role of divergent adaptation in speciation[4,16].

Nevertheless, research combining a clinal perspective with the analytical power of whole-genome sequence data is largely lacking[13]. Here, we present such an investigation in a threespine stickleback fish (*Gasterosteus aculeatus*) population pair residing in parapatry (that is, contiguously) in Misty Lake and its inlet stream (Vancouver Island, British Columbia, Canada)[17,18] (Fig. 1a). This system is relatively young (postglacial, < 10,000 generations old; a generation is 1−2 years) and the populations exhibit no obvious genomic incompatibility when crossed[17–19]. Ecological differences between the lake and stream habitat, however, have driven dramatic genetically-based adaptive divergence in several traits including body shape, breeding coloration, trophic morphology and behavior[17–22] (Figs. 1b and 2a and Supplementary Fig. 1). Low-resolution marker data further indicate that phenotypic divergence between the two populations coincides with substantial genetic differentiation ($F_{ST}$ ~0.12[23–25]). This divergence has almost certainly arisen in the face of gene flow, as opposed to reflecting secondary contact after evolution in isolation. The reason is that temporally stable spatial associations of concurrent phenotypic and genetic discontinuities of different magnitudes with lake-stream habitat transitions are ubiquitous in stickleback[26–29]. By contrast, the conditions for the evolution of neighboring lake and stream populations in initial physical isolation must be rare in general, and appear entirely implausible in watersheds within which divergent lake and stream populations have evolved repeatedly in a spatio-temporal sequence[30]. The small scale of the Misty system in particular makes the initial physical isolation of the lake and inlet stream habitat appear improbable hydro-geographically. Given the absence of physical dispersal barriers and the potential of stickleback to disperse over hundreds of meters in a few days (even against water current[31,32]), potent reproductive barriers between the lake and stream population must exist.

To characterize reproductive isolation between Misty Lake and stream stickleback at the genomic level, we perform a clinal investigation based on pooled whole-genome sequencing at a fine geographic scale. Combined with individual-based simulations tailored to this system, our study offers a fine-grained illustration of how divergent natural selection can drive and maintain reproductive isolation between populations in direct contact.

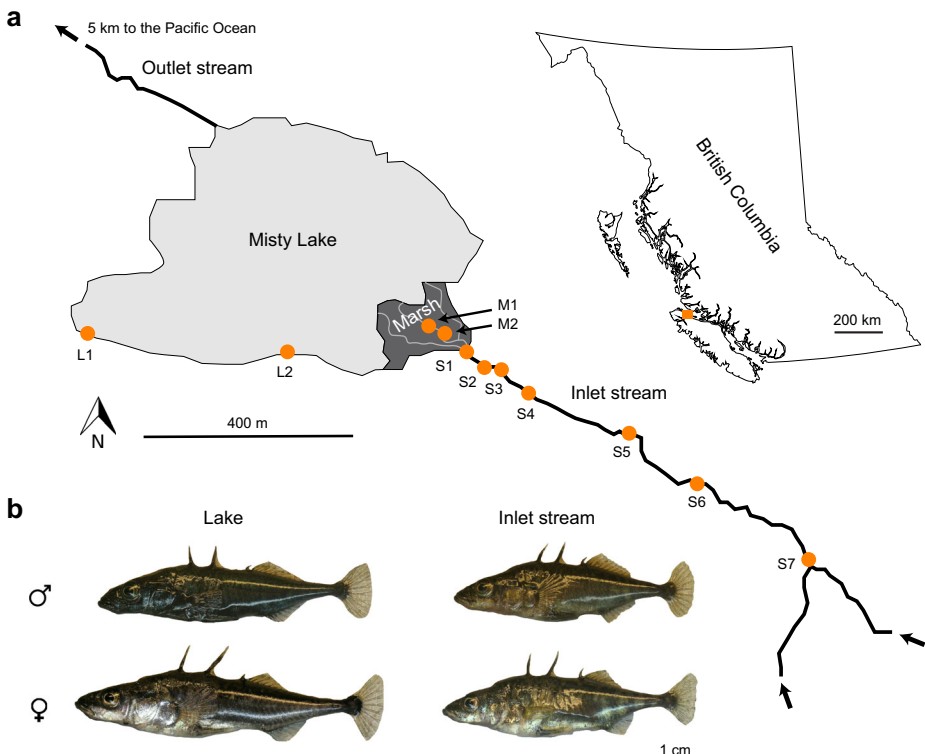

**Fig. 1 The Misty Lake and inlet stream stickleback system. a** Geographic situation of Misty Lake, its inlet stream, and the marsh located between these habitats (map created by the authors based on data from Google Earth). Sample sites along the lake-stream transition are indicated by orange dots (GPS coordinates and sample sizes given in Supplementary Table 1). **b** Representative lake and stream stickleback individuals of both sexes (Photo credit: Katja Räsänen).

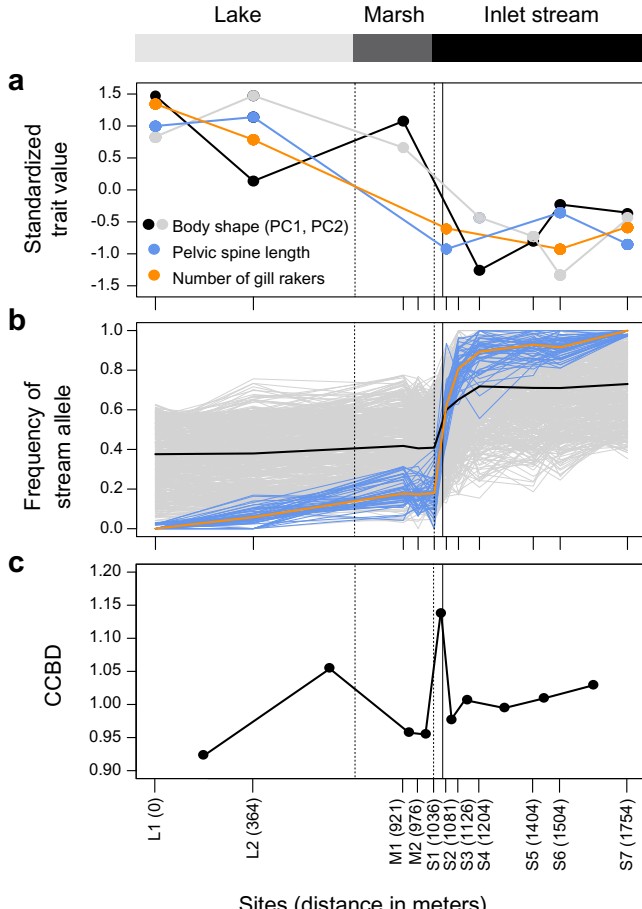

**Fig. 2 Phenotypic and genomic differentiation across the lake-stream transition. a** Morphology, including geometric morphometric body shape (principal component scores, details in Supplementary Fig. 1), a predator defense trait (pelvic spine length), and a foraging trait (gill raker number). For ease of presentation, only site means are shown. Data were available for a subset of the study sites only. **b** Frequency of the stream allele at the 50 independent SNPs exhibiting strong differentiation (AFD ≥ 0.97) between the most distant sites L1 and S7 (i.e. the selected SNPs, blue lines; their median frequency is shown in orange), and at 500 neutral SNPs (gray lines; median frequency in black). Source data are provided as a Source Data file. **c** Chromosome center-biased differentiation (CCBD), calculated for all pairs of neighboring samples. The midpoint between the paired samples is used as location along the gradient. In all panels, the dotted vertical lines indicate the lake-marsh and marsh-stream habitat boundaries and the black vertical line indicates the cline center estimated for the selected and neutral SNPs (1071 m). The distances on the X-axis represent the approximate swimming distance between each site and the lake site L1.

## Results and discussion

**Polygenic adaptive divergence between lake and stream stickleback**. To initiate our investigation, we sampled ~56 stickleback individuals from each of 11 sites across the Misty Lake and inlet stream transition, with a distance of < 2 km between the most distant sites (L1 and S7, Fig. 1a and Supplementary Table 1). Each sample was subjected to pooled whole-genome sequencing to about 100x read depth, yielding ~1.9 million single-nucleotide polymorphisms (SNPs) as genetic markers.

We hypothesized that phenotypic and genetic divergence between the lake and stream fish maintained at a small geographic scale must be tightly linked to divergent selection between the habitats. Our first objective was therefore to identify

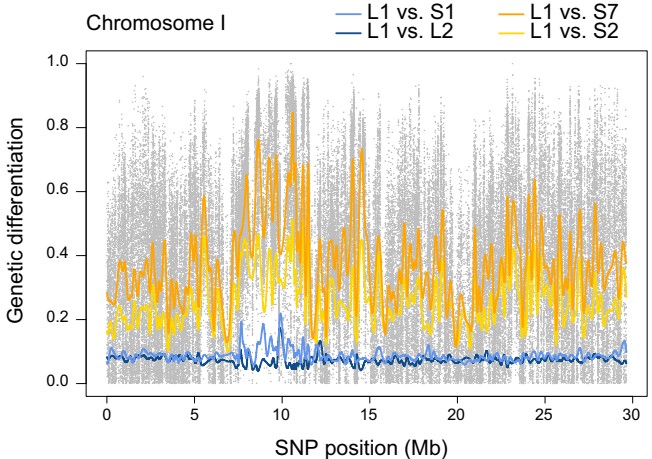

**Fig. 3 Pairwise genetic differentiation between clinal sites along a representative chromosome.** Genetic differentiation is expressed as absolute allele frequency differentiation (AFD) for four site comparisons, each involving the lake site L1. The colored lines represent differentiation smoothed across SNPs by non-parametric regression. For the comparison between the endpoint samples (L1-S7), differentiation is additionally shown for all SNPs individually (gray dots). Note that the differentiation profile from the L1-S1 comparison resembles the within-lake (L1-L2) comparison, except for a few genome regions particularly strongly differentiated in the L1-S1 comparison. The latter also prove exceptionally strongly differentiated between the lake and the more distant stream sites.

genome regions likely targeted by selection by performing a standard pairwise population comparison. We here focused on our two most distant lake and stream sites (L1 and S7, Fig. 1a), assuming that these represented our stickleback samples the least influenced by gene flow, and hence the best adapted to local lake or stream ecology. For this site pair, we quantified the magnitude of genetic differentiation, expressed as the absolute allele frequency difference AFD[33], across all genome-wide SNPs.

This genome scan revealed a median differentiation of 0.35 across all SNPs (expressed as $F_{ST}$: 0.14), thus confirming the strong overall genomic differentiation between the lake and stream population indicated previously by sparser marker data[18,23–25]. However, numerous genome regions exhibited much stronger differentiation between the two most distant samples, and dozens of SNPs proved fixed for alternative alleles (differentiation along a representative chromosome is presented in Fig. 3, and along all chromosomes in Supplementary Fig. 2). We next defined all autosomal SNPs exhibiting AFD ≥ 0.97 in this site comparison as exceptionally highly differentiated ($n = 162$, ~0.01 percent of all autosomal SNPs; genome-wide AFD and $F_{ST}$ distributions are visualized in Supplementary Fig. 3). These SNPs were considered to tag distinct high-differentiation genome regions only if they were separated by at least 50 kb. From each of the independent SNP clusters defined in this way, we then chose one representative high-differentiation SNP at random, yielding our panel of 50 selected SNPs representing putative genomic targets of strong divergent selection. Interrogating the gene annotation of the stickleback genome revealed that only one out of the 50 selected SNPs lay within a coding gene sequence, consistent with primarily regulatory evolution during stickleback diversification[34]. (The same result was obtained when expanding this check to the full 162 high-differentiation SNPs: only seven mapped to coding sequences of four different genes.) Overall, our comparison of a single site pair makes clear that genomic divergence in Misty Lake-stream stickleback is strong and highly

polygenic; divergent selection likely targets hundreds of loci, as inferred in other lake-stream stickleback systems[25,35,36].

**Strong reproductive isolation at the habitat transition.** Having obtained genomic evidence of the general presence of divergent selection, we took advantage of our full spatial set of samples to explore how tightly divergence was related to eco-geography. Graphing the frequency of the stream allele (i.e., the major allele at the site S7) at the selected SNPs across all 11 clinal sites uncovered a dramatic genetic shift over a physical stream distance of merely 45 meters, starting at the transition of the marsh to the stream (blue and orange lines in Fig. 2b; the stream site S1 is located in immediate proximity to this transition). This habitat transition also coincided roughly with a major shift in ecologically important traits, as revealed by morphological data from this and previous studies for a subset of our sites (Fig. 2a; note that the spatial resolution of the phenotypic data around the marsh-stream transition is low). To expand our focus beyond adaptive genetic and phenotypic variation, we next delimited two categories of SNPs unlikely to be physically close to loci under divergent selection, hence reflecting genome-wide background differentiation by drift. These included 500 SNPs chosen at random from all genome-wide autosomal SNPs deviating by no more than 0.1% from the genome-wide median AFD in the L1-S7 comparison (hereafter called our neutral SNPs), and a SNP panel derived analogously based on just half the median AFD (0.175; our loDiff SNPs). Inspecting the frequency of the stream allele (here defined as the allele more frequent in sample S7 than L1) at the neutral and loDiff SNPs revealed genetic clines as sharp as in the selected SNPs, again starting at the marsh-stream transition (gray and black lines in Fig. 2b and Supplementary Fig. 4). To compare the location of the genetic clines among our marker categories more formally, we fitted classical geographic cline models[37] to the stream allele frequencies for each of the 50 selected SNPs, and for a random subset of 50 markers from the neutral and loDiff SNP panels. This indicated similar cline centers among all SNP categories, located at ~1070 m, consistent with our visual inference (Fig. 4 left, Supplementary Fig. 5).

Together, these spatially fine-grained analyses reveal a remarkably tight association between ecology, phenotype, and genome-wide genetic variation. Divergent selection thus not only maintains adaptive divergence, but simultaneously generates strong barriers to gene flow across the entire genome. What are these barriers? A major contribution to reproductive isolation must arise directly from local adaptation, in the form of reduced performance of migrants and hybrids between the habitats[1,2,4,38,39]. Field transplant experiments in a European lake-stream stickleback system indicate that these reproductive barriers alone can reduce lake-stream gene flow by ~80%[40]. Given that phenotypic differentiation in this European lake-stream

system is weak compared to other lake-stream systems[41], selection against migrants and hybrids should cause an even stronger reproductive barrier in the phenotypically highly divergent Misty stickleback. Adaptive divergence may additionally entail some sexual isolation, although experiments in Misty[42] and European[43] lake-stream stickleback indicate that this barrier is relatively weak. Moreover, reproductive isolation across the lake-stream habitat transition is plausibly promoted by habitat preference[31]. However, in no parapatric lake-stream system assessed so far have crossing experiments under standardized laboratory conditions[17–19,36,44] indicated intrinsic inferiority of hybrids (F1, F2) or backcrosses. Our evolutionarily young study system thus offers no support for the idea that steep genetic clines and the underlying strong reproductive isolation reflect primarily intrinsic, ecology-independent genetic incompatibilities having become spatially coupled with clines at loci under (potentially weak) divergent ecological selection[45,46].

**Gene flow is restricted to a narrow zone.** Having uncovered reproductive isolation at the habitat transition, we next asked how strong this isolation is. An informative pattern in our clinal genomic data was that for all three SNP categories (selected, neutral, loDiff), the frequency of the stream allele increased substantially upstream of the marsh-stream transition – but only up to sampling site S4, that is, over c. 150 m (black and gray curves in Fig. 2b and Supplementary Fig. 4). Beyond this location, allele frequencies consistently proved stable across the remaining stream section investigated. (This interpretation does not conflict with a subtle allele frequency shift between S6 and S7, especially at the selected SNPs; this is expected because SNP ascertainment was conditioned on allele frequencies from the site S7 in all three marker categories.) In agreement with visual inference, cline modeling estimated a median cline width of just 127 m for the selected SNPs, although somewhat wider clines were estimated for the neutral and especially the loDiff SNPs (Fig. 4 right and Supplementary Fig. 5). However, we suspected that the latter may represent an artifact of model fitting, which was assessed with simulated data differing exclusively in the magnitude of differentiation between the populations (Supplementary Fig. 6). This confirmed that identically abrupt allele frequency breaks lead to increasing cline width estimates and to increasing spread in cline center and cline width with decreasing AFD between the populations, as observed empirically across our SNP categories (Fig. 4). Our cline modeling thus yields no indication that allele frequency clines are wider in the neutral or loDiff SNPs than in the selected SNPs.

Collectively, our analyses show that beyond a few hundred meters upstream of the marsh-stream transition, reproductive isolation must already be so strong that a homogenizing effect of gene flow from the lake is no longer detectable in any marker category. Misty Lake and stream stickleback thus support models suggesting that ecologically based divergent selection on numerous loci may jointly drive reproductive isolation strong enough to block gene flow across the genome as a whole[1,47–49].

Although stream allele frequencies in all three SNP categories increased substantially over a few hundred meters upstream of the marsh-stream transition, these frequencies remained largely stable downstream of the transition, that is, all the way from site S1 across the marsh to the most distant lake site (Fig. 2b and Supplementary Fig. 4). This indicates asymmetry in gene flow; genetic variation flows predominantly from the lake (and marsh; this habitat will be discussed in a later section) into the lower reach of the stream than in the opposite direction. While this asymmetry may be influenced by differences in dispersal behavior between lake and stream fish, we believe that a main reason is

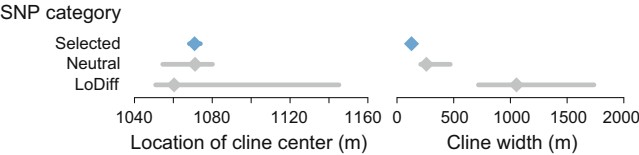

**Fig. 4 Cline parameter estimates for different SNP categories.** Estimation of the geographic location of the cline center and the width of the cline by cline modeling for the selected SNPs (blue; AFD ≥ 0.97 in the L1-S7 comparison), and the neutral (AFD near the genome-wide median) and loDiff (AFD near half the median) SNPs (gray). The diamonds indicate the median value across the 50 independent SNPs in each marker category, and the bars give the associated bootstrap 95% compatibility intervals. Source data are provided as a Source Data file.

imbalance in relative population sizes: according to estimates from mark-recapture data, the lake population is substantially larger than the stream population[50]. Moreover, contrary to the lake, the stream is a linear habitat; a relatively smaller fraction of the total stream population may have the opportunity to disperse into the lake than vice versa[3]. The asymmetry in gene flow provides further evidence that beyond a few hundred meters upstream of the habitat transition, reproductive isolation must be strong across the whole genome. If the latter was not the case, one would expect cline centers to be displaced upstream in the neutral and especially the loDiff SNPs relative to the selected SNPs. However, our cline modeling confirms similar cline center locations among our marker categories (Fig. 4 and Supplementary Figs. 5 and 7).

**Gene flow in the contact zone is heterogeneous across the genome.** The occurrence of gene flow from the lake to the lower reach of the stream allowed us to ask if some genome regions were more strongly isolated by divergent selection than others. A pattern relevant to this question emerged when quantifying genetic divergence by pairwise genomic comparisons of each of the sites L2 to S7 to the lake site L1, and graphing the resulting population differentiation along chromosomes. This analysis revealed that the lowest stream site (S1; located right at the marsh-stream transition) was generally differentiated only trivially from the lake across most of the genome, hence producing chromosomal differentiation profiles closely resembling those from the within-lake (L1-L2) site comparison (compare the light-blue and dark-blue curves in Fig. 3 and Supplementary Fig. 2). Interestingly, however, a small number of genome regions exhibited substantially stronger differentiation in the L1-S1 comparison (e.g., ~10 Mb on chromosome I, Fig. 3; note that owing to the higher sensitivity of AFD to weak population differentiation compared to $F_{ST}$[33], this pattern was easier to discern with the former differentiation metric, Supplementary Fig. 8). This led us to hypothesize that our stream site closest to the lake (S1) was overwhelmed by gene flow from the lake, with appreciable genomic differentiation from the lake maintained in genome regions under the strongest divergent selection only. If true, these specific regions should, compared to randomly chosen genome regions, exhibit exceptionally strong differentiation between the lake and the stream population in general, including between the two samples from the endpoints of our geographic gradient (L1, S7). This prediction was confirmed by simulations estimating the magnitude of L1-S7 differentiation for the two categories of genome regions (i.e., highly differentiated in the L1-S1 comparison and randomly chosen) expected if drift was similar between these regions (Fig. 5). Genome regions particularly highly differentiated between the lake and the closest stream site (S1) thus harbor loci under strong divergent selection that partly resist homogenization by gene flow. We speculate that during the formation of the lake-stream stickleback pair, initial divergence at these loci set the stage for genomically more widespread adaptive divergence that now overall constitutes a strong reproductive barrier between the populations[1,47,49].

While this heterogeneity in gene flow between the populations concerned relatively small genome regions, we also obtained evidence of heterogeneous gene flow at the scale of whole chromosomes. Specifically, stickleback (like eukaryotes in general[51]) exhibit substantially elevated recombination rates near the chromosome peripheries compared to the chromosome centers[52]. Under this recombination rate distribution, theory predicts that polygenic divergent selection with gene flow causes relatively elevated population differentiation in chromosome

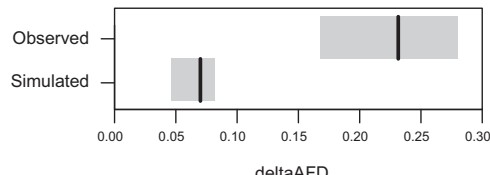

**Fig. 5 High-differentiation regions maintained by divergent selection at the habitat transition.** The upper row represents the empirically observed difference in AFD (deltaAFD) in the L1-S7 site comparison between high-differentiation regions and randomly chosen regions ($n = 178$ each). These regions were identified based on the L1-L2 and L1-S1 comparisons. The lower row represents deltaAFD calculated analogously for simulated high-differentiation and control loci evolving to similar baseline differentiation under drift alone ($n = 178$ each). The black vertical lines indicate median values and gray boxes represent bootstrap 95% compatibility intervals. DeltaAFD observed empirically is much greater than expected under drift alone, indicating that the high-differentiation regions identified in the L1-S1 comparison must be under particularly strong divergent selection between Misty Lake and inlet stream stickleback in general.

centers[53,54]. The reason is that in the chromosome centers, maladaptive foreign chromosome segments recombining into the locally favored genomic background will tend to be longer and hence to harbor a greater number of locally maladaptive alleles, thus making their selective elimination more efficient. We thus quantified the magnitude to which genetic differentiation is elevated across chromosome centers relative to the peripheries (i.e., chromosome center-biased differentiation, CCBD[53,54]) for all pairwise combinations of neighboring samples. This bias proved greatest for the S1-S2 sample comparison (Fig. 2c), that is, at the cline center location estimated for all SNP categories. This supports the notion of an antagonism between selection and gene flow in the lowest reach of the inlet stream.

Collectively, our analyses indicate that reproductive isolation between the Misty Lake and the inlet stream population is very strong, allowing the evolution of the two populations largely unconstrained by gene flow. Nevertheless, the lowest section of the stream represents a zone in which selection is opposed by ongoing gene flow from the lake. This gene flow must involve hybridization between lake and stream fish, not just dispersal of lake fish into the lower stream section. The reason is that both heterogeneous genomic divergence in the L1-S1 sample comparison and the CCBD peak in the S1-S2 comparison require differential lake-stream gene flow among genomic regions, hence hybridization. Nevertheless, investigating in more detail based on individual-level sequence data to what extent haplotypes typical of the lake and stream populations are mixed by recombination within the contact zone before they are eliminated by selection is an exciting future opportunity in this stickleback system.

**Strong reproductive isolation in simulations of polygenic divergence.** We have inferred from empirical patterns that reproductive isolation between parapatric stickleback populations reflects a by-product of adaptive divergence. To support the plausibility of our interpretation theoretically, we tailored individual-based simulations to the Misty stickleback system. We assumed nine demes in a linear array, with dispersal occurring in the beginning of every generation between contiguous demes only (stepping-stone model; Fig. 6a). Considering empirical population size estimates[50], the first (lake) deme was specified to be larger than all other (stream) demes together. The two habitats were under polygenic divergent selection, with fitness being a function of genetic variation at 100 loci. All loci were

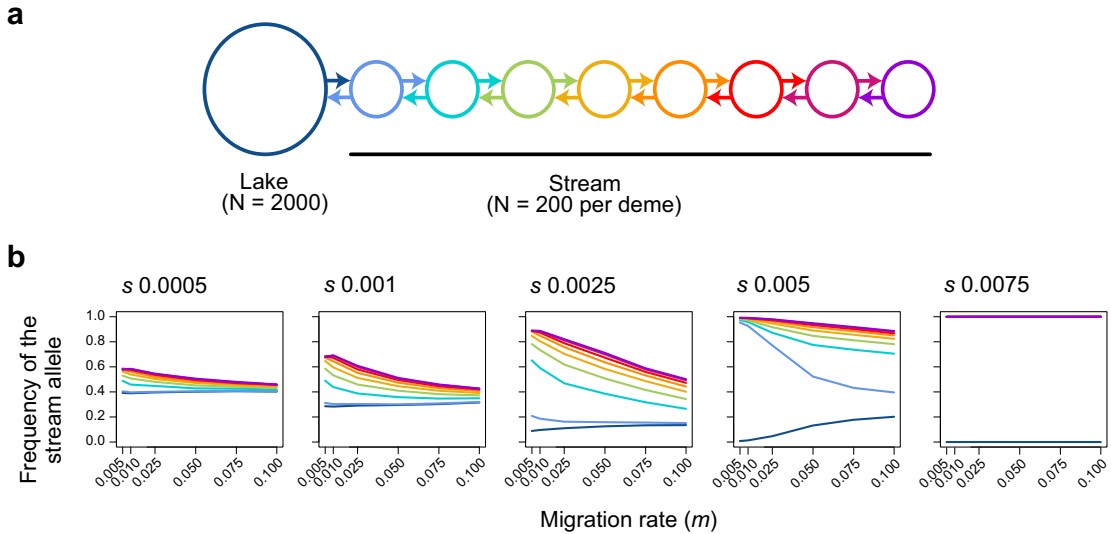

**Fig. 6 Simulated divergence with gene flow across the lake-stream habitat transition. a** Schematic of the stepping-stone model. Arrows indicate migration between neighboring demes. **b** Median frequency of the allele favored in the stream across 100 loci under selection after 1000 generations of evolution, averaged across 20 simulation replications for different combinations of migration rate and selection strength (selection coefficient $s$ given on top of each panel). The demes are color coded as in **a**. In the rightmost panel, all stream demes are fixed for the stream allele hence the lines overlap.

polymorphic in the beginning of the simulations, thus mimicking standing genetic variation well known to underlie adaptive diversification in stickleback[29,34,55–59]. After 1000 generations of evolution, we examined clinal patterns in the frequency of the stream allele (analogously to our empirical analyses of the selected SNPs) resulting from different combinations of dispersal rates between the demes and strengths of divergent selection between the habitats.

A first observation from these simulations was that strong adaptive divergence (and hence necessarily the associated reproductive barriers) established easily across the habitat transition, even for combinations of relatively small selection coefficients and high dispersal rates (Fig. 6b). Specifically, selection coefficients just below 0.01 were already sufficient to allow complete differentiation between the lake and all stream sites across all dispersal rates considered (up to 0.1). Second, we found that whenever gene flow prevented complete lake-stream divergence, the stream site closest to the lake was particularly strongly constrained by gene flow, whereas more distant stream sites showed relatively similar allele frequencies (e.g., Fig. 6b, selection coefficients of 0.0025 or 0.005 combined with relatively high dispersal rates). This pattern closely resembled the shape of the cline in allele frequencies observed upstream of the marsh-stream transition in all SNP categories (Fig. 2b and Supplementary Fig. 4). All these observations remained qualitatively consistent across several robustness checks (Supplementary Fig. 9).

The simulations support our empirically based conclusion that adaptive divergence from abundant standing genetic variation has produced strong reproductive isolation in the absence of physical barriers. Our study thus provides insights into the genomic architecture of adaptive divergence: previous theory has emphasized that in the face of gene flow, adaptive divergence is promoted by the physical linkage of adaptive alleles, as produced by inversions[60–62]. However, in Misty Lake-stream stickleback, we found no indication of divergence in inversion polymorphisms. Importantly, the three large inversions often involved in adaptive divergence between pelagic and benthic stickleback

ecotypes[29,34,55,57] proved monomorphic across our samples. Not denying the importance of chromosomal rearrangements in adaptive divergence in some study organisms, our whole-genome clinal investigation highlights that polygenic selection per se, without any particular physical arrangement of the targeted loci, can be sufficient for the emergence of strong divergence and reproductive isolation in the face of gene flow[47,49].

**Perturbation of gene flow-selection balance by an unusual ecological event.** Our clinal phenotypic data and allele frequencies in all SNP categories revealed that the stickleback inhabiting the marsh (sites M1 and M2) are genetically very similar to the true lake fish (L1, L2). Nevertheless, in a previous study, microsatellite loci designed to discriminate Misty Lake and inlet stream fish indicated hybridization in the marsh[22]. Similarly, the frequency of the stream allele at our selected SNPs was slightly elevated in the marsh relative to the lake samples (although this may be attributable to the ascertainment of these SNPs, see above). This raised the possibility that the marsh might allow a modest degree of genetic mixing between the lake and the stream population despite being strongly dominated by lake fish. If true, we hypothesized that changes in the level of dispersal from the lake or the stream into the marsh, as mediated by a physical perturbation of the system, should drive a measurable shift in the genetic composition of the marsh fish.

Evaluating this hypothesis was made possible by exceptionally intense rainfall during our main sampling period, causing an unusual rise in inlet stream discharge and lake water level that for a few days flooded the marsh that normally is above water level (Supplementary Fig. 10). To assess the genomic consequences of this event, we complemented our standard marsh sample (M1) taken before the flood by two additional samples from the same site, taken during the flood and 1 year later. The comparison of these temporal samples at lake-stream population-distinctive SNPs (AFD ≥ 0.75 in the lake pool [sites L1 and L2 combined] to stream pool [S6 and S7] comparison) revealed a striking decline (often to zero) of the stream allele frequency during the flood, that is, within a few days (Fig. 7). Although our pooled sequence

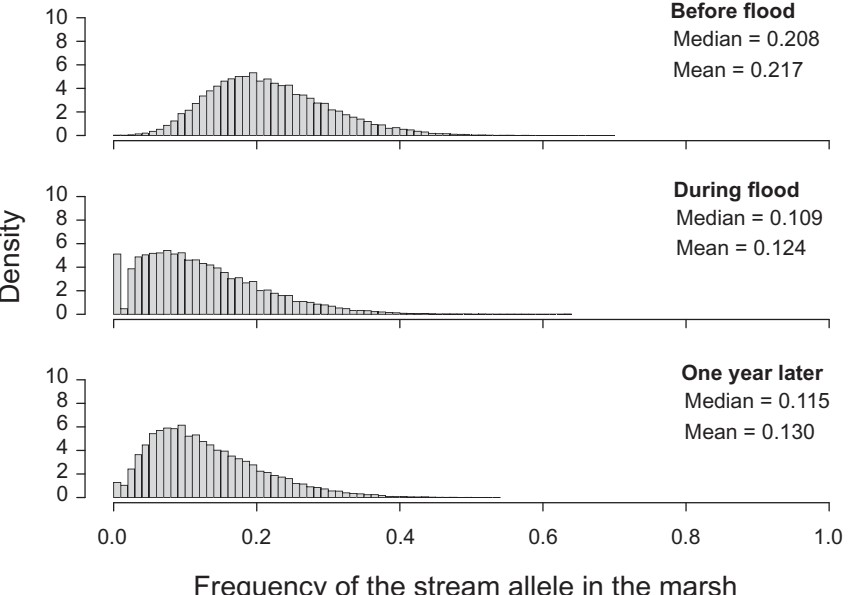

**Fig. 7 Impact of a flood on the genetic composition of stickleback in the marsh.** Shown is the distribution of the frequency of the stream allele across 49,677 SNPs exhibiting strong lake-stream differentiation (AFD ≥ 0.75 in the comparison between the lake pool [sites L1 and L2 combined] and the stream pool [sites S6 and S7 combined]) in stickleback sampled at the marsh site M1 at three different time points. The median and mean of the distributions are also given. Source data are provided as a Source Data file.

data precluded inspecting haplotype structure, the speed of this genetic change clearly indicates extensive dispersal of lake fish into the marsh, facilitated by easier access to the latter habitat. This conclusion is supported by simulations suggesting that at least 90% of the stickleback residing at the marsh site before the flood must have been replaced by migrants from the lake during the flood (Supplementary Fig. 11). Interestingly, this perturbation in allele frequencies at the marsh site appeared partly offset one year later (e.g., the number of SNPs monomorphic for the lake allele declined by 76%; Fig. 7), indicating selection against the lake migrants and/or the new immigration of stream individuals from the nearby contact zone.

Our genomic analysis of temporally replicated samples from the marsh supports the idea that stickleback in this habitat represent a genomic mix between the lake and stream population caused by hybridization[22]. Nevertheless, genetic material from the lake population vastly predominates in the marsh, and we demonstrate that this imbalance can become nearly complete temporarily by an unusual short-term ecological modification of dispersal opportunities. Further disentangling the relative importance of selection and gene flow as determinants of allele frequencies within this eco-geographically intermediate habitat will benefit from direct information on the local selective conditions and individual-level genomic sequence data.

To summarize, our investigation of parapatric stickleback has demonstrated a tight link between ecology, polygenic adaptive divergence, and whole-genome reproductive isolation, thereby illustrating how adaptation and speciation can be two sides of the same coin. Genetic exchange between the diverging populations, however, has not ceased completely but continues within a narrow contact zone. We show that the balance between homogenizing gene flow and divergent selection in this zone is fragile and can shift quickly when habitats are perturbed. Our work highlights the power of whole-genome sequencing at a fine spatial scale and across multiple time points to inform the eco-geography and genetic architecture of speciation.

## Methods

**Stickleback sampling and phenotypic analysis.** Stickleback were captured with unbaited minnow traps at 11 sites in Misty Lake and its inlet stream between May and July 2016, during the breeding season (the marsh site M1 was additionally sampled in August 2017). Sample sizes ranged from 40 to 62 individuals per site (details on the locations and samples given in Supplementary Table 1). From each individual, a dorsal spine was clipped and stored in 95% ethanol for DNA extraction. All individuals were immediately released.

To allow qualitatively linking genomic to phenotypic differentiation along the geographic gradient, we performed a geometric morphometric body shape analysis. For this, 40 individuals from a subset of our sites (L1, L2, M1, S4, S5, S6, and S7) were photographed on their left side with a standard scale using a digital camera (Canon PowerShot G11, Canon, Tokyo, Japan). All photographs were digitized with tpsDIG2 (life.bio.sunysb.edu/morph/) by the same investigator (KBO) in haphazard order by placing 14 landmarks used in previous studies in the same system[24,63]. Using the geomorph R package[64,65], the resulting coordinates were aligned and generalized Procrustes analysis performed, yielding principal components of body shape variation among individuals[64,66]. In addition, we retrieved data for two ecologically important traits related to predator defense (pelvic spine length) and foraging (number of gill rakers) from another subset of our sites (L1, L2, S2, S6, and S7) studied in a previous phenotypic study of Misty Lake and inlet stickleback[67]. All these phenotypic data were mean-centered and standardized to allow visualization on the same scale. All animal work in this study was conducted in accordance with the Animal Use Protocol from McGill University.

**DNA library preparation and sequencing.** DNA was extracted individually from the dorsal spine clip of each of the 701 total stickleback using the Quick-DNA TM Miniprep Plus Kit (Zymo Research, Irvine, CA, USA), following the manufacturer protocol. For enzymatic tissue digestion, the spines were minced with micro spring scissors to maximize DNA yield. Following DNA quantification using a Qubit fluorometer (Invitrogen, Thermo Fisher Scientific, Wilmington, DE, USA), individuals were pooled to equal molarity without PCR-enrichment to obtain a single DNA library per sampling site (and per time point in the case of the marsh site M1). The 13 total libraries were paired-end sequenced to 151 base pairs on 10 total lanes of an Illumina HighSeq2500 instrument, producing a median read depth per base pair of 103x across the libraries (min = 51, max = 133; Supplementary Table 1). Combined with the relatively large number of individuals per site, this high read depth is expected to allow estimating allele frequencies with high precision[68].

**SNP discovery.** Raw sequences reads were parsed by sampling site and aligned to the third-generation assembly[69] of the 447 Mb stickleback reference genome[34] by using Novoalign (http://www.novocraft.com/products/novoalign/; settings given in the Supplementary Software). The Rsamtools R package[65,70] was then used to convert the alignments to BAM format, and to perform base counts at all genome-

wide positions for each sample using the *pileup* function. To identify informative single-nucleotide polymorphisms (SNPs), we first combined nucleotide counts from the two lake samples (L1 and L2) and from the two most upstream inlet samples (S6 and S7) into lake and stream pools. Genomic positions then qualified as SNPs for further analysis if they exhibited a read depth between 50 and 400x within each pool (to exclude poorly sequenced and repeated regions), and a minor allele frequency (MAF) of at least 0.25 across the two pools combined (to ensure a high information content[71]). Throughout the study, allele frequencies were calculated directly from raw nucleotide counts. The 1,920,596 SNPs passing these filters were genotyped in all 13 samples separately.

**Quantifying clinal genomic differentiation**. Genetic differentiation along the lake-stream gradient was quantified by two approaches. The first relied on the frequency of the stream allele at selected, neutral, and loDiff SNPs. The selected SNP category comprised markers showing genetic differentiation ≥ 0.97 between the geographically most distant samples (L1 and S7). Throughout the study, we quantified genetic differentiation by the absolute allele frequency difference AFD[33], although a few key analyses were repeated by using $F_{ST}$. We considered only nuclear markers, and ignored the sex chromosome (XIX) because it was enriched for high-differentiation SNPs relative to the autosomes, as expected from its reduced population size and hence stronger drift. Including the sex chromosome, however, always had a trivial influence on the results. If multiple high-differentiation SNPs were < 50 kb apart, they were considered a cluster from which only one SNP was chosen at random to ensure statistical independence, resulting in a panel of 50 independent selected SNPs. As a resource for future investigations, we retrieved from the reference genome annotation all genes located within a 50 kb window centered at each of the selected SNPs, producing a gene compilation (provided as Supplementary Data 1) likely containing numerous genes under divergent lake-stream selection in the Misty system. The annotation was also used to assess if the selected SNPs were located within or outside coding gene sequences. The neutral SNPs, in turn, represented 500 markers chosen at random among all autosomal SNPs for which AFD deviated from the genome-wide median differentiation in the L1-S7 comparison (c. 0.35) by no more than 0.1%, again applying a 50 kb spacing threshold. The loDiff SNPs, finally, represented markers weakly differentiated relative to the genome-wide median; they were chosen analogously to the neutral SNPs, but targeting an AFD deviation of 0.1% around just half the genome-wide median differentiation (c. 0.175). At all selected, neutral, and loDiff SNPs, we then defined the nucleotide relatively more frequent in the S7 than the L1 sample as the stream allele. Finally, the frequency of the stream alleles was calculated for each sample and visualized along the geographic gradient. In the second approach, we calculated genetic differentiation at all genome-wide SNPs for each pairwise combination of the first lake sample (L1) and all other samples, requiring a read depth between 50 and 200x within each sample. The values obtained were visualized along chromosomes, raw and/or smoothed by non-parametric regression using the *smooth.spline* R function (band width 0.1). Genome-wide median differentiation for these site comparisons, and for all comparisons between neighboring sites, is given in Supplementary Fig. 12, expressed as both AFD and $F_{ST}$.

**Cline modeling**. As a numerical complement to our visual cline analyses, we fit our allele frequency data to classical geographic cline models implemented in the HZAR R package[37]. We here used the sampling site-specific frequencies of the stream allele, the total nucleotide counts underlying these frequencies, and the geographic locations as input data, set allele frequency intervals to the observed maximum values, and assumed two independent tails (models without tails produced qualitatively similar results). We considered all 50 selected SNPs for modeling, and random subsets of the same size from the neutral and loDiff SNPs. For each SNP, cline fitting was run in 10 replicates, and the median maximum likelihood estimate of cline center and cline width across these replicates was recorded. We then compared these key cline features among the SNP categories based on the median values across SNPs and the associated 95% bootstrap compatibility intervals (10,000 resamples). The raw data distributions are provided in Supplementary Fig. 5. Because the ascertainment of the SNPs in all three categories was contingent on the magnitude of differentiation between our most distant sites (L1, S7), thus generating subtle allele frequency shifts between each of these sites and their adjacent site (most pronounced in the selected SNPs; Fig. 2b), we repeated all cline modeling by excluding the sites L1 and S7. This produced qualitatively similar results leading to the same conclusion (Supplementary Fig. 7a, b).

Our visual analysis revealed stable allele frequencies over several hundred meters in the upper reach of the stream for all three SNP categories (Fig. 2b). By contrast, cline modeling indicated an increase of cline width from the selected to the neutral and loDiff SNPs. This discrepancy led us to hypothesize that the inverse relationship between cline width and AFD among the SNP categories may be a modeling artifact, which was confirmed by simulation (details presented in Supplementary Fig. 6).

**Inferring selection-gene flow antagonism from high-differentiation regions**. While inspecting the comparison L1-S1, we observed that some genome regions showed remarkably strong genetic differentiation compared to the overall

undifferentiated genome-wide background. This led us to speculate that in these specific regions, genetic variants particularly strongly favored in the stream were maintained at elevated frequency at the S1 site, while the remainder of the genome was overwhelmed by gene flow from the lake. Assuming particularly strong divergent selection on these genome regions, we predicted that they should exhibit particularly strong differentiation between the lake and the stream population in general, including in the comparison L1-S7. To evaluate this idea, we first subtracted the mean AFD value in the L1-L2 comparison (considered the differentiation baseline within the lake habitat) from the corresponding value in the L1-S1 comparison for each genome-wide 10 kb sliding window (5 kb overlap) containing at least 5 SNPs. For the most highly differentiated 0.5% of these windows (high-differentiation windows, HDW; $n = 178$; median AFD difference between the comparisons L1-S1 and L1-L2: 0.132), considered regions under strong divergent lake-stream selection, and for the same number of windows chosen at random as control (non-HDW; median AFD difference: 0.011), we then calculated mean AFD in the L1-S7 comparison.

Because the HDW by definition exhibited elevated differentiation in the L1-S1 comparison, somewhat stronger L1-S7 differentiation in these windows relative to random windows was expected even if the HDW were strongly differentiated in the former comparison just by chance. Comparing the two categories of windows thus required a benchmark, which was obtained by individual-based simulation. We here constructed $n$ haploid individuals by first generating 178 biallelic (1, 0) loci (non-HDL) at which the stream allele (1) occurred at a frequency specified by a random draw from the uniform distribution bounded between 0.05 and 0.5 (i.e., the stream allele was always the minor allele, as observed empirically at the site S1; Fig. 2b). Another set of loci (HDL) was generated analogously, except that the frequency of the stream allele was increased by 0.1, corresponding to the observed difference in median L1-S1 AFD between the HDW and non-HDW. We then allowed this population to evolve neutrally (i.e., to drift) by drawing offspring for each new generation at random with replacement. All loci were unlinked, and random assortment of alleles was achieved by swapping alleles between the haplotypes within pairs of offspring. After $g$ generations, we determined median AFD before vs. after evolution for the HDL and non-HDL. The combination of $n$ and $g$ was chosen to produce drift at the non-HDL approximating the median AFD observed across the non-HDW in the L1-S7 comparison ($n = 200$, $g = 1000$; higher values for both variables produced similar results but required longer simulation). This simulation was replicated 25 times.

Finally, we calculated the difference in median AFD in the L1-S7 comparison between the empirical HDW and non-HDW, and the median difference in AFD achieved during simulated evolution at the HDL and non-HDL across the 25 replicates, both referred to as deltaAFD. Uncertainty around these point estimates was obtained by bootstrapping windows (empirical data) and replicates (simulated data) 10,000 times. Elevated empirical relative to simulated deltaAFD would indicate that regions exhibiting the strongest differentiation in the L1-S1 comparison also show exceptionally strong differentiation in the L1-S7 comparison relative to the genome-wide baseline, consistent with these regions being targets of particularly strong divergent selection between lake and stream stickleback.

**Inferring selection-gene flow antagonism from CCBD**. The combination of polygenic selection, gene flow, and a reduced crossover rate in chromosome centers compared to chromosome peripheries causes relatively elevated population differentiation in chromosome centers (chromosome center-biased differentiation, CCBD[51–54]). To explore the strength of selection-gene flow antagonism along the geographic gradient, we thus quantified to what extent genetic differentiation was biased toward chromosome centers for all 10 pairwise comparisons of neighboring samples. For this, we defined the outer 5 Mb on either side of a chromosome as high-crossover rate periphery and the remainder of the chromosome as low-crossover rate center[52]. Next, we divided median AFD across all central SNPs by median AFD across the peripheral SNPs for each chromosome within each site pair. For each pair, we then treated the median across these ratios as CCBD, and graphed this metric along the lake-stream gradient by using the midpoint between the neighboring samples as geographic location. As a robustness check, this analysis was repeated by using as site pairs each of the samples L2 to S7 combined with L1, which produced very similar results supporting the same conclusion (Supplementary Fig. 13).

**Individual-based simulations of divergence with gene flow**. To explore theoretically how selection can drive and maintain reproductive isolation in the presence of gene flow, we conducted individual-based forward simulations using a diploid stepping-stone expansion of the model in Berner and Roesti[54]. Our standard model involved nine total demes arranged in a one-dimensional array, with adjacent demes connected by migration (Fig. 6a). The first deme ($n = 2000$) represented the (larger) lake population while all other demes (each $n = 200$) represented stream sites. In the beginning of each generation, a fraction of $m$ individuals was chosen at random from each deme to migrate into the neighboring deme on either side (juvenile migration). A total fraction of $2m$ thus emigrated from each deme, except for the demes located at the endpoints of the array, for which this fraction was $m$. The migration phase was followed by selection and reproduction. We modeled polygenic divergent selection by assuming a total of 100 biallelic unlinked loci under divergent selection, with one allele favored in the first

deme and the other allele favored in all other demes. The simulations started with the frequency of both alleles set to 0.5 in all demes, to minimize the probability of the stochastic loss of adaptive variation[54]. Loci contributed additively to fitness; each maladaptive allele reduced an individual's fitness from the local fitness optimum of one by $s$, the selection coefficient. Individual fitness was then scaled by the mean population fitness and determined an individual's probability to be drawn for reproduction. Individuals reproduced as hermaphrodites and were allowed to mate more than once, each mating producing one offspring. Mating was repeated until the number of offspring re-established initial local deme size. The offspring cohort then replaced the parental deme (discrete generations) and entered the migration phase.

We explored a range of combinations of migration rates (0.005−0.1) and selection coefficients (0.0005−0.0075). All simulations were run for 1000 generations, a time span shown by preliminary runs over up to 7000 generations to allow approaching migration-selection balance (Supplementary Fig. 14a, b). All parameter combinations were replicated 20 times. Results were visualized by plotting for each deme at generation 1000 the mean across simulation replications of the median frequency of the allele favored in the stream across all 100 loci for all combinations of migration rates and selection coefficients.

The robustness of the standard model described above was assessed by a number of additional simulations, presented in Supplementary Fig. 9. We here considered models with a multiplicative (as opposed to additive) contribution of each locus to overall fitness; a lower number of selected loci (10); physical linkage among all loci by assuming a single chromosome undergoing uniformly distributed crossover during mating (as opposed to independent segregation); locus-specific selection coefficients drawn at random from the exponential distribution with a rate equal to $1/s$ (as opposed to identical selection coefficients among loci); and greater population size imbalance by setting the lake deme ten times (as opposed to 1.25x) larger ($n = 10{,}000$) than all stream demes combined ($n = 125$ per deme).

**Quantifying the impact of a flood on stickleback in the marsh.** For the marsh site (M1), three temporally replicated samples were available: before, during, and one year after a flood. The former represents the sample also used in all previous analyses; the latter two samples were processed in exactly the same way. To maximize the sensitivity for detecting gene flow, we here considered only SNPs highly differentiated (AFD ≥ 0.75) between the lake pool (sites L1 and L2 combined) and the stream pool (sites S6 and S7), and sequenced to a minimum read depth of 50x within each temporal sample. For the 49,677 SNPs thus obtained, we visually compared the frequency of the stream allele among the samples.

Because this analysis indicated massive dispersal of lake stickleback into the marsh during the flood, we explored by simulation how much of such dispersal was needed to drive the observed change in allele frequencies. For the SNPs above, we here averaged allele frequency data from the marsh before the flood and those from the nearest lake site (L2), considering a wide range of relative proportions of the latter (10%−100%). Comparing visually the resulting (mixed) allele frequency distributions to the one observed during the flood allowed a qualitative assessment of the proportion of lake dispersers into the marsh during the flood. This proportion was additionally explored using Approximate Bayesian Computation (ABC) (details given in Supplementary Fig. 11). Unless stated otherwise, all our analyses and simulations were implemented in the R language[65].

**Reporting summary**. Further information on research design is available in the Nature Research Reporting Summary linked to this article.

## Data availability
All raw Illumina sequences, demultiplexed by site (and sampling period for the site M1) are available from the European Nucleotide Archive (accession numbers ERS4388731-ERS4388743) under the project PRJEB37366. Raw genome-wide nucleotide counts for all sites (and temporal replicates) are provided on Dryad (https://doi.org/10.5061/dryad.c59zw3r67). Source data are provided with this paper.

## Code availability
Codes are provided as Supplementary Software.

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

## Acknowledgements
This project was supported financially by the Swiss National Science Foundation (grant 31003A_165826 to DB) and the Freiwillige Akademische Gesellschaft Basel (QH), and field sampling additionally by Fisheries and Oceans Canada, the British Columbia Ministry of the Environment, and the McGill University Biology Department. We thank Fiona Beaty, Brody Forst, Bailey Feddersen, Tristan Kosciuch, Minxin Lu, Erica MacClaren, Emily McIntosh, Alexanne Oke, Sarah Sanderson, and Mac Willing for aiding field sampling; Western Forest Products for providing logistical and safety support and access to field sites; Walter Salzburger for sharing web lab infrastructure; Brigitte Aeschbach and Nicolas Boileau for facilitating lab work; Christian Beisel, Ina Nissen and Elodie Burcklen for Illumina sequencing at the Quantitative Genomics Facility, D-BSSE, ETH Zürich; the developers of Novocraft for sharing their sequence aligner; Katja Räsänen for providing pictures of Misty stickleback; Laurent Guerard and Nicolás Lichilín Ortiz for help with scripting. Computation was performed at the sciCORE (https://scicore.unibas.ch) scientific computing center of the University of Basel.

## Author contributions
D.B. initiated and supervised the study; D.B. and Q.H. designed the experiment; D.B., Q.H., and A.P.H acquired funding; K.B.O. and A.P.H. performed field sampling and measurements; K.B.O. generated and analyzed phenotypic data; Q.H. and T.G.L. performed wet lab work; D.B. and Q.H. implemented analytical tools; Q.H. and D.B. analyzed genomic data and interpreted results; Q.H. and D.B. wrote the manuscript, with input from all co-authors.

## Competing interests
The authors declare no competing interests.
