## [Peer Review File · Nature Communications]

Reviewers' Comments:

Reviewer #1:

Remarks to the Author:

Haenel et al describe genome-wide allele frequency clines along an ecological gradient (stream to lake) in stickleback fish. They document numerous SNPs fixed (or nearly fixed) for alternative alleles over a short geographic distance (<1km). These fixed SNPs are spread across the genome suggesting numerous regions of the genome are resistant to gene flow. They interpret this as evidence of a genome-wide barrier to gene flow (I generally agree, but see below). This is a nice contrast to (the more numerous) studies showing free genetic exchange between taxa across most of the genome with only a few regions (barrier loci) resistant to gene flow. And it is particularly striking given the spatial scale and potential for gene flow. As such, I think this is an important study that advances our understanding of the evolution of strong barriers to gene flow and thus the completion of speciation.

My main criticism of the paper is that this aspect of the study is not more fully developed. In particular:

1. Theory developed by Nick Barton and others shows how linkage disequilibrium (LD) among barrier loci can increase in a tension zone resulting in a positive feed-back loop leading towards greater indirect selection and LD creating a strong barrier to gene flow (via coupling, and the related concept of genome-wide congealing developed by Sam Flaxman and colleagues). Some of this literature is cited, but not really unpacked and discussed. I think this paper would benefit from being placed in the context of the coupling hypothesis (or at least by discussion this hypothesis and empirical results relevant for it more thoroughly). To me, this is broader context for the work than genome-scale data (which are becoming increasingly common and are arguably not used to the fullest extent in this paper anyway... which is fine).

2. I am mostly happy with the analyses presented, but I think an opportunity was missed by not fitting formal, multilocus cline models. This should be done for the fixed SNPs and a subset of random SNPs (e.g., the 500 random SNPs). The reason being that, given a tension zone model (which the authors suggest applies here), multilocus cline models provide estimates of the relative strength of direct (based on the exponential tails of the clines) and indirect selection (that is, selection caused by the build-up of LD among barrier loci and between barrier and non-barrier loci, which can be inferred from the width of the central component of the cline). Such estimates should of course be interpreted with due caution, but they provide a means to formally quantify the extent to which patterns of introgression are consistent with an overall, genome-wide barrier to gene flow versus direct selection on individual loci. This gets at the heart of the goals of the study in my opinion.

Other minor comments and suggestions (with line numbers):

L24, I don't feel strongly about this, but my preference is to restrict the term polygenic to describing trait genetic architecture rather than patterns of genetic differentiation (one could instead say genomically widespread or something similar). The authors should feel free to disregard my preference on this if they wish.

L121-125, I have mixed feelings about representing each of the 34 regions by a single SNP. This might be useful for some analyses as it allows one to think of the 34 SNPs retained as (mostly) independent, but in other cases, such as considering genome annotations or simply describing patterns of introgression, using all fixed SNPs makes more sense (and seems more complete). At minimum, this choice should be better justified and the authors should at least consider showing clines for all fixed SNPs in a supplementary figure.

L167-170, It isn't clear to me why the authors are dismissive of sexual isolation and habitat preference. Even if a small subset of genes were associated with these traits, if they prevent the formation of hybrids in the wild they could contribute substantially to a genome-wide barrier to gene flow. Maybe there is something more about the biology of this system that I don't know that makes this unlikely.

L417, I don't see a description of how allele frequencies were estimated. Perhaps simply from the allele counts? More sophisticated methods that take sequence quality scores into account and capture uncertainty in allele frequencies exist (though using such approaches probably won't alter the conclusions of this paper in a meaningful way).

L421, Should 0.25 be 0.025? If not, this is a rather extreme minor allele frequency cut-off and should be justified (I am quite happy with 0.025).

Zach Gompert

Reviewer #2:

Remarks to the Author:

In this manuscript, the authors use pooled whole genome resequencing to investigate adaptive divergence and speciation between lake-stream stickleback morphs in the Misty Lake system on Vancouver Island, British Columbia. The authors take advantage of the ability to sample clinally in the system, sequencing pools from 11 different sample sites spanning the lake, marsh and stream habitats. Using the extremes of their clinal transect, they identify candidate SNPs fixed between the lake and the stream in order to identify genome regions likely under strong divergent selection. They then visually examine the clinal shifts in frequency of these alleles along the lake-stream transect and draw comparisons with concordant clinal variation in phenotypic traits too. The authors conclude this is the result of strong reproductive isolation in a narrow tension zone between the lake and stream habitats. They further support this firstly with a series of pairwise comparisons of genome regions under apparent divergent selection at the tension zone and between the extremes of the lake-stream system. The authors also incorporate individual simulations in order to investigate the parameter space that might lead to adaptive divergence over a habitat transition. Finally, the study is concluded with a temporal study in the marsh tension zone of the Misty system where an extreme rainfall event altered dispersal and caused changes in the stream allele frequency distribution

There is a lot of impressive work in this manuscript and I really enjoyed reviewing it. Generally, it is well written and clear and there is little doubt that the subject matter and the study system are of interest. The authors make a good point that clinal analysis is not widely used enough with genomic data and that there is a lot of potential for incorporating cline theory into speciation genomics. Furthermore, the authors have done a great job of incorporating empirical, simulation data and even essentially a natural experiment with the rainfall perturbation. These components all have the makings of a really good paper – however as I explain below, more work is necessary to push this MS over that line.

In my opinion the manuscript has a number of quite substantial issues, perhaps the most glaring of all being that despite promoting the use of cline analysis, there is no inclusion of cline theory or formal clinal analysis at all. This is a real shame as clearly there is great power here to actually quantify some of the parameters discussed throughout the MS (selection coefficients, dispersal, cline centre). This is also coupled with a lack of clarity over methods, particularly with regards to genomic regions putatively involved in reproductive isolation and the omission or sweeping over of some important key points from the recent speciation genomics literature (i.e. background selection, recombination rate variation, structural variation). Furthermore, the authors make a number of extremely strong statements about their results without clear support (i.e. the claim that reproductive isolation is near complete – LN244 and elsewhere)

Taken together, I do ultimately think that this manuscript would be of value for Nature Communications. There is little doubt that the study has been well-designed and implemented. However, in my opinion the MS requires a substantial reframing and a stronger justification. I think the authors are on the right track with regards to relating their work to an increasing body of literature on implementing clinal analyses with genomic data, it would just help their case more to actually do that. Below I flesh out my major comments in more detail and also provide some minor

comments on clarification and adjustments in the text.

Include a formal clinal analysis – as I've already mentioned, one thing that immediately struck me when reading this MS is that it is largely justified by highlighting the fact that clinal variation is overlooked in recent speciation genomics literature. This is true and the authors also highlight a number of studies where it has been applied. However, this is odds with the fact that the MS really contains no formal cline analysis at all – only a graphical analysis (Fig 2) of clinal variation in allele frequencies. Cline theory is not straightforward and there are only a handful of packages which can handle cline data, and those are probably not fine-tuned for pool seq data. Nonetheless, there have been some clear attempts to incorporate cline theory and genomic data – i.e. Westram et al 2018 *Evol Letters*. Furthermore, that study uses individual-based simulations (much like the ones applied here) to determine the threshold of significance of cline parameters. Given the wealth of data from the Misty system, including estimates of dispersal and census population size, incorporating a more thorough cline analysis would greatly strengthen the paper.

Better recognition of potential bias from recombination rate variation – a key aspect I felt missing from the MS was the influence of recombination rate variation on the probability of a genome region being identified as an outlier. This is an important issue, particular in sticklebacks as there is evidence of bias towards low recombination regions (i.e. Samuk et al 2017 *Mol Ecol*). It would be interesting to see how the probability of a region containing a fixed SNP varies in this way and could easily be modelled with a logistic regression or something similar. There is certainly some attempt to address this, particularly with CCBD and the short discussion of chromosome centres but I think this would be strengthened with a more formal test involving recombination rate, a quantitative measure of crossover rate.

Overstating the case for complete reproductive isolation – there are several points in the MS where the authors make some strong suggestions that reproductive isolation is near complete in the Misty system (i.e. LN181; LN244). This is an overinterpretation of the results. It is in fact possible to qualitatively estimate the extent of RI from hybrid zones using hybrid index but of course, the use of pool-seq data here means that is not possible. The authors have not actually measured reproductive isolation in this case – they are instead examining allele frequency differences between populations. I certainly agree with them that there is good evidence for strong divergent selection in this system and that this has shaped a strong cline in allele frequencies based on genome regions which might underlie morphological and ecological divergence. However, allele frequencies can be influenced by other processes too (i.e. linked selection, drift) and furthermore, allele frequency differences alone are poor estimators of effective migration rate (that is, reproductive isolation at the level of the genome – see Whitlock & McCauley 1999). A more appropriate interpretation of the pattern of increase in stream allele frequency is that there is stronger selection against the dispersal of lake alleles with the increase in distance from the tension zone. It is worth noting here that my earlier suggestion of a formal estimation of cline parameters for loci might also give some indication of how different genome regions differ in their dispersal from the cline centre, which in turn might lead to a more insightful analysis with regards to gene flow and introgression across this habitat transition.

Minor comments

LN47-48: It is quite a strong statement to say that comparisons between ecologically divergent species largely ignores the eco-geography of speciation. On the contrary, there are many examples where this is an important aspect of the experimental design of these studies. Martin et al 2013 is a good example of this, where eco-geography is explicitly used to look at different levels of differentiation between species. Similarly, contact zones have long been recognised as areas of importance for understanding how RI translates to the genomic level. So, while I agree clinal analysis has not been well used in the early years of the genomic era, this is clearly being rectified now (as is seen by the references cited in LN51). I think this justification for the study needs to be reframed.

Fig 4 and the L1-S1 comparison – not really clear. I understand the method on rereading but the rationale is not obvious – probably needs better explaining

LN65: Paraptry – typo.

LN69: What does genetically compatible mean? Presumably no evidence of sterility or effect on viability. I think it is better to say no obvious genomic incompatibilities.

LN77: Given the context of this system and lake-stream stickleback in general as a model for parallel (or non-parallel) evolution, it's probably better to avoid the word parallel here. It confused me on first glance!

LN82: Another example of a need to be careful with a word here. Effective could cause confusion with a term such as effective migration (which is actually very relevant in this case as clearly there is a high probability for migration of individuals, the question is whether their genes migrate too). I would suggest replacing this.

LN96: Name the sites for clarity. Are these distant stream or lake sites? Or between the lake and the stream (i.e. as in LN105).

LN111: For the sake of comparison, citing the F_{st} value would also be helpful. This is largely for others reading the study and to make it at least partially comparable to other genomic data (i.e. for example – the work of some of the authors!)

LN116: what determines a region? From Fig 3 it looks like there are only 3 SNPs which are actually fixed. I don't doubt the evidence for high levels of differentiation, but are the regions defined arbitrarily? Otherwise, why not use a method such as a HMM in order to identify regions of elevated AFD?

LN128: Without a proper example of what the authors mean by regions, it is very hard to actually assess whether this is the case – again, what is a differentiated region?

LN151: Here again, the word effective can be confusing. Do the authors mean that the barriers work? I think it is important to be careful with the wording used to ensure the correct interpretation of the results.

LN165: Inviability – however these are F1 crosses right? What about in backcrosses or F2s? Also inviability is only one component of incompatibilities – has hybrid sterility been checked?

LN173: I don't think you've really shown reproductive isolation – you've demonstrated evidence of genetic differentiation which points towards a level of RI here. I think this needs to be softened.

LN180: I do not follow the logic here – this does not mean RI is complete. Selection against introgressed stream alleles means they cannot move beyond S4 in the contact zone.

LN215: Substantially stronger – but not really justified other than geographically – what is the height of this peak and the actual difference in AFD between the two comparisons?

LN220: Why not also look for regions

LN238: CCBD needs better explaining here.

LN244-247: This is really overstated – it is not clear that reproductive isolation is complete here and you have not actually directly measured RI – thus you cannot state this.

LN279: "Were sufficient"

LN298-300: What about other inversions though? Is there any evidence of this from the current data? It is quite possible that other inversions segregate in lake-stream fish

LN312-320: Hybridization between lake and stream fish could also drive the more extreme PC1, PC2 body shape differences compared to L2. Transgressive segregation among hybridising lineages can drive this – although it is difficult to test for.

LN23-324: The flood event is an interesting addition to the paper – how often do such floods occur? Is it likely they are semi-regular events that would introduce migrants into the marsh environment? If so, I think it is worth emphasising the role of ecological/environmental cycles on passive migration – perhaps often overlooked in the literature with regards to dispersal between environments

LN338-339: Is this greater than expected by chance though? This could probably be simulated, given the estimates of the proportion of fish migrating into the marsh (i.e. Supp Mat Fig 5).

LN432 & 435: Why 100Kb? Is this where LD decays to the background rate in stickleback?

LN886-887: I may have missed this but what qualifies as strong lake-stream differentiation here? Is it the same as LN569-571? In which case, this should also be explained in the results & discussion.

Data sharing – why not share the allele frequency data too? And the genotype calls?

Reviewer #3:

Remarks to the Author:

In their manuscript entitled “Clinal genomic analysis reveals strong reproductive isolation across a steep habitat transition in stickleback fish” Haenel and co-authors focused on a well-characterized stickleback population pair adapted to lake and stream habitats in Misty river, with the main goal of understanding (at the genomic level) how adaptive divergence results in strong reproductive isolation. The implementation of a whole-genome pool-seq approach to characterize multiple populations across an environmental transect from lake to stream habitats, complemented by the analysis of phenotypic traits, revealed steep clines of allele frequencies for multiple markers across the genome, with cline centres roughly coinciding with the habitat transition. This pattern, combined with individual-based simulations, was interpreted as evidence for strong polygenic adaptive divergence forming a genome-wide barrier to gene flow without physical isolation.

The study of a contact zone at this level of resolution (both geographic and genomic) offers an unprecedented power for understanding speciation at different levels (from genotype to phenotype). Thus, I think it can be of wide interest for researchers working on many different systems in the field. While I found this study powerful and highly interesting, I have a number of significant concerns that are important to consider.

i) Although I found the manuscript well-written, I think that the central question was not very clear. By the end of the introduction, the authors mention that this study aims to identifying reproductive barriers between lake and stream sticklebacks (in the Misty river system) at the genomic level. However, the impact of the different reproductive barrier types on gene flow was not assessed, and the analysis of SNP clines alone can hardly allow to identify which traits or components of isolation contribute as barriers to gene flow.

ii) I understand the constraints in terms of space but it would be important to have some more background about the system. For instance, are we dealing with a primary or secondary contact? Another important aspect concerns stickleback migration. Do they migrate to breed in this lake/stream system? If they do, it would be important to know when was sampling conducted relative to breeding migration. Finally, some reproductive barriers in this system and their putative influence in the observed patterns should be described in more detail. In particular, the authors refer that sexual selection and habitat preference are unlikely to cause massive differentiation across small distances but they do not explain why. In general, I think that habitat choice is one way in which clines can be steep relative to dispersal, generating differentiation at many loci across the genome.

iii) My main criticism to this study is that despite the suggestion (abstract and introduction) that a formal cline analysis will be performed, I was surprised that this was not the case. I am not saying that plotting AFD across the transect is not at all informative. However, there is a body of theory

and tools explicitly developed to extract rigorous information from the type of data the authors collected, such as: quantify how many loci present a significant cline and their distribution across the genome, quantify barrier strength, test differences in cline centres and widths, test for asymmetrical clines, etc. Cline analyses would also allow to explicitly test specific claims about polygenic adaptation (where stepped clines are expected), about allelic frequencies in S1, and about temporal instability of frequencies after environmental perturbation.

iv) There are other methodological aspects would benefit from further improvements: the phenotypic data does not complete match the genetic sampling and is not much used in the analyses, which significantly weakens some inferences; it is not clear why the authors only used 500 random snps representing the non-fixed differences (even for these not much detail is given about the variability in clines and how are they distributed across the genome); it would be useful to know the uncertainty (error distribution) around the allele frequencies estimates from pool-seq data; there is no information about recombination rate, despite the fine-scale maps available for stickleback. Finally, although I am fully aware of the problems related with FST, I am also not entirely sure that AFD (Berner, 2019) is free of biases. Would not be useful to compare the main results using AFD vs FST?

v) Given the effects of recombination (CCBD analysis), I would like to hear from the authors about the possibility of some of the AFD peaks resulting from background selection. We know that this system is relatively young and probably not many mutations occurred since divergence; it is known from previous studies that there is divergent selection in thus system; and the cline centres seem to be located at the habitat transitions. However, I was wondering if some of the peaks (mainly in the central regions of chromosomes where recombination is lower) could have resulted from background selection. Gene flow could have erased its signatures in the extremities of chromosomes close to the habitat transition where gene flow is higher but it might not be enough to counteract it in the centre of the chromosomes. This may not be strongly plausible but I would like to know the authors opinion about this alternative interpretation.

vi) Concerning the simulations, I think that the drift-alone model is not an appropriate null because it is known to be selection. The genome-wide effects of the barrier to gene flow can originate clinal patterns at sites that are unlinked to selected loci (see for example Westram et al. (2018) for a similar approach). The authors mention that their empirical results are compatible with polygenic selection they modelled but no alternative models (other than drift-alone) are tested and there was no formal comparison to the observed data. The core simulation results reported are focused on the model with unlinked loci, but the chromosome centre effect shows that recombination matters. The authors mention that they additionally simulated physical linkage among all loci along a single chromosome with recombination but it is not clear what is the LD between loci, how LD decays with distance and why a single chromosome. Since the authors likely have access to genetic maps and know the marker positions, I think it would be important that their "standard simulation models" incorporate recombination and to say more about the distribution of clines across the genome to support their claims about a genome-wide barrier.

All in all, I think the system is quite interesting and can be very informative but I think the data should have been analysed in a more thoroughly manner to support the authors' main claims. Currently, I think the support for sentences such as: "This strong polygenic adaptive divergence must constitute a genome-wide barrier to gene flow because a steep cline in allele frequencies is observed across the entire genome..."; "Our simulations support our empirically-based conclusion that adaptive divergence from abundant standing genetic variation has produced strong reproductive isolation in the absence of physical barriers"; and "Collectively, our analyses indicate that reproductive isolation between the Misty Lake and the inlet stream population as a whole is nearly complete, allowing the evolution of the two populations largely unconstrained by gene flow." is rather weak. In fact, it is not clear to me why the authors conclude that reproductive isolation in this system is nearly complete. If we study populations far enough from the contact zone, there will likely be substantial differentiation even if selection is weak. How much differentiation also depends on time and population size. Once more, I think the cline analyses would be useful to measure the barrier strength.

Minor comments:

L22. "Dense" is vague and non-informative.
L23. Same for "numerous".
L44. Remove "genetic".
L45. They can also harbour loci involved in incompatibilities.
L60. Why only initiating?
L76-78. Can the authors present some numbers?
L84. Replace "Mistly" by "Misty".
L92. Is "Results and Discussion" together as single section a valid format for Nature Communications?
L97. I think the authors should indicate here the number of samples used in each pool (average and range).
L116. Are they fully independent? Does LD decays in less than 100kb in this system?
L128. Not clear how the authors go from 34 to hundreds. Could not the target of selection be a single gene (or regulatory) per region?
L133. "action" instead of "presence".
L138. Is not it actually a little bit shifted to the stream side? Is there any explanation for this shift?
L139-142. It is hard to see that it coincides. The shape of the clines is very different from the fixed SNPs. I think this needs to be properly tested before the authors conclude that it coincides.
L144. Why only 500 SNPs? Please clarify if these can be located within the 34 highly differentiated genomic regions (even if they are not the 34 fixed SNPs).
L146. But the difference between their frequencies at both ends of the transect are much different.
L167-170. Not clear why they are unlikely to cause strong differentiation across small distances. Please explain. For example, why habitat preference cannot cause a strong barrier between lake-stream stickleback.
L183. Not clear why "entire" is in italics. Same in L245 for "as a whole"
L259-262. Can long distance dispersal occur in this system, even if at low frequencies?
L338-340. I see some changes in the distribution of allele frequencies but I think that is far from being offset, even if partially. Should not this be tested statistically? Additionally, I think that the use of tools like Admixture would be important to understand if there was actually some admixture to claim gene flow (L31) or simply immigration without gene flow.
L395. Remove "by"
L473. Why haploid individuals were simulated?
L482. Why were all loci unlinked?
L542. Why not males and females?
L543. Why one offspring? Is this usual in sticklebacks?
L551. I found the number of replicates for these and the previous simulations to be relatively low. Can the authors somehow show information about the variance among replicates?
L559. How physical linkage was modelled?
L848. I am not sure we can call it strongly differentiated mainly when compared with differentiation between L1 and other S sites.
L886. How strong?
Figure 5b. It is not clear which demes are fixed for the stream allele.

Reviewer #4:

Remarks to the Author:

This study combines genomic data, morphological analyses, and simulations to demonstrate a strong polygenic basis of reproductive isolation between adjacent lake, marsh and stream populations. Specifically, this study used clinal analyses of phenotype and genetic variation along an habitat transition, using both morphometric analyses and pooled whole genome resequencing data. These empirical analyses are supported by spatially explicit genetic simulations of adjacent demes with different rates of gene flow and selection. Additionally, a natural event (flood) that occurred during the study provides support for the authors' conclusions. Overall I enjoyed reading this paper, and think that with a few revisions, it will be a useful contribution to the literature. Below are some suggestions.

General comments:

- Pooled WGS data was used, presumably to enable whole genome sequencing for many

individuals, but pooled data does limit inferences in some ways and these should be discussed. In particular, I am thinking of the difficulty of identifying admixed or hybrid individuals from pooled data, and how that might limit understanding of this system, specifically as relates to the frequency and fitness of hybrids.

- Claims of adaptive divergence require strong evidence that there are indeed fitness consequences to not having the putatively adaptive phenotype. As far as I can tell, there are no direct observations of fitness (of hybrid or translocated individuals) in this particular system. European examples of transplant/hybrid fitness are cited, but it is unclear how applicable they are to this system, which differs from other stickleback systems in several ways (e.g., in the polygenic arrangement of differentiation and absence of large inversions associated with ecotypes). Is there evidence beyond the coincidence of phenotypic, ecological, and genetic clines in this system for the increased fitness of individuals with putatively adaptive traits? Without direct evidence of selection, inferring a tension zone is potentially difficult, and a more cautious presentation of this inference might be warranted.

- AFD, allele frequency difference, is used as the primary measure of genetic divergence between lake and stream fish. While I think this is a valid choice, I would like to see some explanation of why AFD is used instead of *F_{st}*, which is more common (yes, I know there's a whole paper about this and it's cited in this manuscript, but a brief justification would be helpful in this manuscript). I am also curious about whether results would look different if *F_{st}* was used instead of AFD - does *F_{st}* miss some of the differentiation, or result in different inferences? Is there a methodological point to be made here too for researchers working in other low-divergence systems or incipient species about AFD vs *F_{st}*?

Specific items:

- Units of coverage are unclear (Line 403) - is it a mean of 103 reads per locus for pools of ~50 individuals, or 103 reads per locus per individual? I suspect it is the former, but these are very different levels of coverage, so clarification is needed here.

-HDW and HDL are not defined explicitly at first use (Line 464, 474), although it is possible to infer meaning from the surrounding text. Please add definitions the first time you use these acronyms to make this clearer.

Minor comments/typos:

Line 65 - "Parapatry" where it should be "parapatry"

Line 84 - "Mistly" where it should be "Misty"

Line 264-266 - awkward phrasing, sentence fragment

Complete reviewer comments (in black) and our responses (in blue)

Reviewer #1 (Remarks to the Author):

Haenel et al describe genome-wide allele frequency clines along an ecological gradient (stream to lake) in stickleback fish. They document numerous SNPs fixed (or nearly fixed) for alternative alleles over a short geographic distance (<1km). These fixed SNPs are spread across the genome suggesting numerous regions of the genome are resistant to gene flow. They interpret this as evidence of a genome-wide barrier to gene flow (I generally agree, but see below). This is a nice contrast to (the more numerous) studies showing free genetic exchange between taxa across most of the genome with only a few regions (barrier loci) resistant to gene flow. And it is particularly striking given the spatial scale and potential for gene flow. As such, I think this is an important study that advances our understanding of the evolution of strong barriers to gene flow and thus the completion of speciation.

My main criticism of the paper is that this aspect of the study is not more fully developed. In particular:

1. Theory developed by Nick Barton and others shows how linkage disequilibrium (LD) among barrier loci can increase in a tension zone resulting in a positive feed-back loop leading towards greater indirect selection and LD creating a strong barrier to gene flow (via coupling, and the related concept of genome-wide congealing developed by Sam Flaxman and colleagues). Some of this literature is cited, but not really unpacked and discussed. I think this paper would benefit from being placed in the context of the coupling hypothesis (or at least by discussion this hypothesis and empirical results relevant for it more thoroughly). To me, this is broader context for the work than genome-scale data (which are becoming increasingly common and are arguably not used to the fullest extent in this paper anyway... which is fine).

1) We appreciate R1's suggestion to link our results better to theory; as we detail below, we have done so. However, we feel first some clarification is needed: R1 here raises two terms – 'coupling' and 'genome congealing'. As we understand theory, these are distinct ideas. Coupling specifically refers to the spatial alignment of *intrinsic (or endogenous), ecology-independent* incompatibilities reducing hybrid fitness with clines at loci under ecological selection, or simply with spatial discontinuities. That is, the coupling model assumes a major contribution to reproductive isolation (RI) by BDM-type of incompatibilities. This is perhaps expressed most clearly in the well-known Bierne et al. 2011 Mol. Ecol. paper on the 'coupling hypothesis': 'local adaptation explains where genetic breaks are positioned, but not necessarily their existence, which can be best explained by endogenous incompatibilities'. Earlier work by Barton makes the same assumption (e.g., Barton 1979; Barton & Hewitt 1985; Barton & De Cara 2009). With its emphasis on intrinsic incompatibilities, the coupling model is relatively unimportant to our study; as we describe on Lines 176-178, intrinsic genetic incompatibilities have never been detected in lake-stream stickleback despite explicit tests. Even without this evidence, most readers would probably understand intuitively that the role of BDIMs can be ignored in populations diverging from shared standing genetic variation for just a few thousand generations. Nevertheless, to address R1's suggestion, we have added a

sentence (L178-183) making clear that our results do not support the coupling model (we here explicitly use the term ‘coupled’) and providing two key references (Barton & De Cara 2009; Bierne et al. 2011).

The second concept in question posits that ecologically based divergent selection cumulated across *numerous* loci may be strong enough to restrict gene flow effectively not only at the selected loci, but across the genome as a whole (Barton & Bengtsson 1986). One proposed nuance in this process is that adaptive divergence may first occur only at the most strongly selected loci, and that the resulting partial RI then facilitates even more adaptive divergence and RI – hence positive feedback (Barton 1983, and particularly Rice & Hostert 1993). Another nuance is that there may be a critical number of loci beyond which genome-wide RI suddenly emerges, hence ‘genome congealing’ (Flaxman et al. 2014, Mol. Ecol.). All this literature is indeed relevant to our study, since we demonstrate strong genome-wide RI driven by polygenic selection, and we agree that it was insufficiently acknowledged. Following R1’s recommendation, we now put our findings in the context of these in the passages inferring strong RI (L211-214) and heterogeneous gene flow (L266-269; here we explicitly mention the feedback process), and we cite key references (Barton, Flaxman).

However, for several reasons, we prefer not to use the polygenic RI / congealing framework as the leading focus in the Introduction. First, our study uncovers *patterns* consistent with these concepts, but does not investigate the proposed underlying *processes* directly; we have little information about the strength of selection on differentiated loci, their initial allele frequencies, or the temporal sequence of their divergence. Second, the predictions regarding divergence under polygenic selection are model-specific – their generality is uncertain. For instance, Barton 1983 predicts a critical value of total selection marking a sharp transition from loci driving RI independently to RI being genome-wide. However, a simulation test of Barton’s analytical theory failed to reproduce such a critical threshold (Baird 1995, Evolution). Likewise, in the Flaxman 2014 congealing simulations, genetic variation arises exclusively from *de novo* mutation, which is different from our system fueled by standing variation. It is thus not clear to us to what extent theory makes predictions beyond the intuitive idea that strong polygenic selection can drive strong (or complete) genome-wide RI, which in our view is essentially the long-standing ‘genomic island’ vs. ‘genome-wide divergence’ topic. Third, although RI is particularly strong in our compared to other ‘ecological speciation’ systems, the ideas related to RI driven by polygenic selection in our view apply to almost *any* genomic study of divergence with gene flow. Finally, our finding of essentially complete ecology-driven RI over such a small spatial distance was a surprise; selling our study as an *a priori* test of any specific genomic divergence model would appear, and indeed be, contrived.

2. I am mostly happy with the analyses presented, but I think an opportunity was missed by not fitting formal, multilocus cline models. This should be done for the fixed SNPs and a subset of random SNPs (e.g., the 500 random SNPs). The reason being that, given a tension zone model (which the authors suggest applies here), multilocus cline models provide estimates of the relative strength of direct (based on the exponential tails of the clines) and indirect selection (that is, selection caused by the build-up of LD among barrier loci and between barrier and non-barrier loci, which can be inferred from the width of the central component of the cline). Such estimates should of course be interpreted with due caution, but they provide a means to formally quantify the extent to

which patterns of introgression are consistent with an overall, genome-wide barrier to gene flow versus direct selection on individual loci. This gets at the heart of the goals of the study in my opinion.

2) This comment echoes similar input by R2 and R3, so it seems analyzing our clinal data by formal modeling in addition to visual inference was generally desired. We have therefore carried out an extensive cline modeling analysis, now presented on L152-158, L196-199 and L512-539 and supported by the new main Fig. 4 and three Supplementary items (Supplementary Figures 5-7). Our interest here was to compare cline centers and widths among SNPs from qualitatively different genome regions (i.e., under divergent selection vs 'neutral'), as suggested by R1 (and R2). Briefly, the idea was that if gene flow from the lake into the stream is more extensive in neutral than selected genome regions, markers tagging the former regions should exhibit a more upstream cline center (since in this system gene flow is asymmetric) and wider clines than the SNPs tagging regions under selection. (Note that our visual inference provided no evidence of such heterogeneity in gene flow, but instead indicated that RI is genome-wide beyond a narrow contact zone.) To make this analysis really powerful, we re-defined our marker panels for the whole study, now using three categories: 'selected' SNPs exhibiting AFD between the terminal lake and stream sites ≥ 0.97 (c. the top 0.0001 of all SNPs), thus certainly tagging DNA segments under divergent selection (results obtained with these SNPs were identical to those obtained with the previous panel of 'fixed' SNPs, but sample size is better in the new panel); the 'neutral' SNPs were chosen to lie tightly around the genome-wide median differentiation (i.e., unlikely to be linked to loci under substantial divergent selection, hence nicely capturing genome-wide drift), whereas the 'loDiff' SNPs were chosen close to just half the median (here an influence of divergent selection can arguably be excluded; this marker category should also mainly mirror evolution by drift). Site-specific allele frequency data for these SNP categories were then fitted to classical cline models implemented in HZAR (Derryberry et al. 2014). This software seems to be state of the art (e.g., Westram et al. 2018 *Evol. Lett.*), and works with data from poolSeq. To summarize the results: no matter what marker category we used, cline center location estimates were identical, and we also found no indication of different cline widths. The conclusion is clear: beyond a few hundred meters from the contact zone, reproductive isolation between lake and stream stickleback is essentially complete genome-wide. Of course, we have also replaced our initial cline graphics by their analogues based on the new SNP categories, and these new graphs lead to the same conclusion (Fig. 2b, Supplementary Figure 4).

Interestingly, while performing this modeling, we discovered an artifact in cline width estimation and confirmed this by simulation (Supplementary Figure 6): in short, cline models return wider clines for markers exhibiting weaker population differentiation, even if the true cline width is held constant. This is worrisome as it may produce misleading conclusions regarding gene flow when using cline fitting software (we qualitatively reproduced this problem with another software; Hlest, Fitzpatrick 2013). We think this artifact has not been recognized previously, so in addition to substantiating our visual inference, the new part on cline modeling makes a relevant methodological contribution.

The reviewers also mention the possibility of interpreting cline tail estimates, or using cline parameters to estimate locus-specific selection strength or dispersal distance etc. We prefer not to do this – given the strong assumptions one must make for such calculations and the problems with reliable cline parameter estimation mentioned above,

we doubt that such numerical exercises would produce biologically meaningful quantities.

To summarize, we have now performed formal cline modeling as requested by reviewers, and the results are fully consistent with our previous conclusions based on visual inference.

Other minor comments and suggestions (with line numbers):

L24, I don't feel strongly about this, but my preference is to restrict the term polygenic to describing trait genetic architecture rather than patterns of genetic differentiation (one could instead say genomically widespread or something similar). The authors should feel free to disregard my preference on this if they wish.

3) We appreciate R1's suggestion, but prefer to stick with the original wording. The reason is that our group has already used this expression widely in the past (see e.g. the article title of Laurentino et al. 2020, Nat. Commun.), and this terminology issue was never raised.

L121-125, I have mixed feelings about representing each of the 34 regions by a single SNP. This might be useful for some analyses as it allows one to think of the 34 SNPs retained as (mostly) independent, but in other cases, such as considering genome annotations or simply describing patterns of introgression, using all fixed SNPs makes more sense (and seems more complete). At minimum, this choice should be better justified and the authors should at least consider showing clines for all fixed SNPs in a supplementary figure.

4) From a study design perspective, enforcing independence among high-differentiation SNPs by applying a minimum spacing threshold between SNPs is uncontroversial in our view. For illustration: among the 72 total SNPs fixed for opposed alleles between our ascertainment samples (L1, S7), ten are on chromosome 13. Inspecting these ten SNPs shows that they are tightly clustered across 4 kb in total, with inter-SNP distances of 2-1794 bp. Plausibly, all these SNPs tag the same selected site (i.e., strong hitchhiking). By contrast, we find only a single fixed SNP on the chromosomes 5 and 16. Accepting ALL fixed SNPs (unfiltered) for analysis would thus both represent classical pseudo-replication and cause results to be influenced strongly by a few genome regions. We feel readers will generally appreciate our spacing filtering – we have followed the same convention in several past studies (e.g., Bisseger et al. 2020 Mol. Ecol., Laurentino et al. 2020 Nat. Commun.). Nevertheless, we have added a brief justification statement to the Methods (L482-484). We have also re-plotted allele frequency clines using unfiltered data, as recommended by R1, and obtain patterns undistinguishable from the analysis with filtered data (we do not present this check, however, because it appears flawed statistically for the above reasons). But of course, we use the unfiltered data for presenting chromosome-wide differentiation (Supplementary Figure 2). As to gene annotation (Supplementary Table 2), we here present all genes within a 50 kb window, so even if some of our SNPs representing high-differentiation regions missed the true target of selection by a few hundred or thousand bases, this would hardly be relevant to this highly exploratory resource.

As a side note: we have now inspected the distribution of high-differentiation SNP spacing along chromosomes in more detail, which showed that 50 kb spacing is sufficient for ensuring high marker independence, so the revised study uses this new

value as threshold (same as in Laurentino et al. 2020 Nat. Commun.). Also, as noted in our response 2 above, we no longer use just the 'fixed' SNPs, but now use 'selected' SNP based on $AFD \geq 0.97$ as filter threshold to have slightly larger sample size (this decision makes absolutely no difference though).

L167-170, It isn't clear to me why the authors are dismissive of sexual isolation and habitat preference. Even if a small subset of genes were associated with these traits, if they prevent the formation of hybrids in the wild they could contribute substantially to a genome-wide barrier to gene flow. Maybe there is something more about the biology of this system that I don't know that makes this unlikely.

5) Ok, our discussion of these additional pre-mating barriers was not sufficiently precise. We have expanded to express that there is good evidence that sexual isolation is likely a weak player, but that a substantial contribution by habitat preference is plausible (L171-176).

L417, I don't see a description of how allele frequencies were estimated. Perhaps simply from the allele counts? More sophisticated methods that take sequence quality scores into account and capture uncertainty in allele frequencies exist (though using such approaches probably won't alter the conclusions of this paper in a meaningful way).

6) Yes, we derived allele frequencies directly from nucleotide counts. Given the large number of individuals sampled per study site (N around 56) and fairly deep coverage (103x on average), our frequency estimates are expected to be excellent, making sophisticated methods unnecessary. As our approach was not clear to R1, we have added an explicit statement (L467-468).

L421, Should 0.25 be 0.025? If not, this is a rather extreme minor allele frequency cut-off and should be justified (I am quite happy with 0.025).

7) Yes, our MAF threshold is 0.25 across the ascertainment population pool. There is a F1000-recommended paper from the last author's lab (Roesti et al. 2012, BMC Evol. Biol.) explaining that when using a low MAF threshold, one accepts markers lacking adequate potential to capture population differentiation, which is undesirable in the present analytical context. This article is well-known, we have used similarly stringent MAF thresholds in all our previous genomic work (e.g., Roesti et al 2012 Mol. Ecol, Roesti et al. 2015 Nat. Commun.; Haenel et al. 2019 Evol. Lett.), no other reviewer raises this issue, and we struggle hard with the reference limit imposed by the journal. We hope R1 understands that we prefer not to justify the MAF threshold in this study again.

Zach Gompert

Reviewer #2 (Remarks to the Author):

In this manuscript, the authors use pooled whole genome resequencing to investigate adaptive divergence and speciation between lake-stream stickleback morphs in the Misty Lake system on Vancouver Island, British Columbia. The authors take advantage of the ability to sample clinally in the system, sequencing pools from 11 different sample sites spanning the lake, marsh and stream habitats. Using the extremes of their clinal

transect, they identify candidate SNPs fixed between the lake and the stream in order to identify genome regions likely under strong divergent selection. They then visually examine the clinal shifts in frequency of these alleles along the lake-stream transect and draw comparisons with concordant clinal variation in phenotypic traits too. The authors conclude this is the result of strong reproductive isolation in a narrow tension zone between the lake and stream habitats. They further support this firstly with a series of pairwise comparisons of genome regions under apparent divergent selection at the tension zone and between the extremes of the lake-stream system. The authors also incorporate individual simulations in order to investigate the parameter space that might lead to adaptive divergence over a habitat transition. Finally, the study is concluded with a temporal study in the marsh tension zone of the Misty system where an extreme rainfall event altered dispersal and caused changes in the stream allele frequency distribution

There is a lot of impressive work in this manuscript and I really enjoyed reviewing it. Generally, it is well written and clear and there is little doubt that the subject matter and the study system are of interest. The authors make a good point that clinal analysis is not widely used enough with genomic data and that there is a lot of potential for incorporating cline theory into speciation genomics. Furthermore, the authors have done a great job of incorporating empirical, simulation data and even essentially a natural experiment with the rainfall perturbation. These components all have the makings of a really good paper – however as I explain below, more work is necessary to push this MS over that line.

In my opinion the manuscript has a number of quite substantial issues, perhaps the most glaring of all being that despite promoting the use of cline analysis, there is no inclusion of cline theory or formal clinal analysis at all. This is a real shame as clearly there is great power here to actually quantify some of the parameters discussed throughout the MS (selection coefficients, dispersal, cline centre). This is also coupled with a lack of clarity over methods, particularly with regards to genomic regions putatively involved in reproductive isolation and the omission or sweeping over of some important key points from the recent speciation genomics literature (i.e. background selection, recombination rate variation, structural variation). Furthermore, the authors make a number of extremely strong statements about their results without clear support (i.e. the claim that reproductive isolation is near complete – LN244 and elsewhere)

Taken together, I do ultimately think that this manuscript would be of value for Nature Communications. There is little doubt that the study has been well-designed and implemented. However, in my opinion the MS requires a substantial reframing and a stronger justification. I think the authors are on the right track with regards to relating their work to an increasing body of literature on implementing clinal analyses with genomic data, it would just help their case more to actually do that. Below I flesh out my major comments in more detail and also provide some minor comments on clarification and adjustments in the text.

Include a formal clinal analysis – as I've already mentioned, one thing that immediately struck me when reading this MS is that it is largely justified by highlighting the fact that clinal variation is overlooked in recent speciation genomics literature. This is true and the

authors also highlight a number of studies where it has been applied. However, this is odds with the fact that the MS really contains no formal cline analysis at all – only a graphical analysis (Fig 2) of clinal variation in allele frequencies. Cline theory is not straightforward and there are only a handful of packages which can handle cline data, and those are probably not fine-tuned for pool seq data. Nonetheless, there have been some clear attempts to incorporate cline theory and genomic data – i.e. Westram et al 2018 *Evol Letters*. Furthermore, that study uses individual-based simulations (much like the ones applied here) to determine the threshold of significance of cline parameters. Given the wealth of data from the Misty system, including estimates of dispersal and census population size, incorporating a more thorough cline analysis would greatly strengthen the paper.

8) Our study now includes a formal cline modeling analysis, as described in our response 2 above.

Better recognition of potential bias from recombination rate variation – a key aspect I felt missing from the MS was the influence of recombination rate variation on the probability of a genome region being identified as an outlier. This is an important issue, particular in sticklebacks as there is evidence of bias towards low recombination regions (i.e. Samuk et al 2017 *Mol Ecol*). It would be interesting to see how the probability of a region containing a fixed SNP varies in this way and could easily be modeled with a logistic regression or something similar. There is certainly some attempt to address this, particularly with CCBD and the short discussion of chromosome centres but I think this would be strengthened with a more formal test involving recombination rate, a quantitative measure of crossover rate.

9) We appreciate this comment, but we are surprised: the very first demonstration of the chromosome-scale impact of heterogeneous recombination rate on population divergence (i.e., CCBD) and outlier detection was made by the last author's research team – in stickleback (Roesti et al. 2012 *Mol. Ecol.*). The same team also published the first characterization of the crossover landscape in stickleback (Roesti et al. 2013 *Mol. Ecol.*), and later a comprehensive theory paper devoted to the role of heterogeneous recombination in divergence with gene flow (and outlier detection), again supported by empirical data from stickleback (Berner & Roesti 2017 *Mol. Ecol.*). Finally, we published a well-cited meta-analysis on heterogeneous recombination across eukaryotes, highlighting how this leads to variation in linked selection and heterogeneous differentiation/diversity across genomes (Haenel et al. 2018 *Mol. Ecol.*).

So the reason why we do not talk a lot about recombination in the present manuscript is certainly not that we overlooked this factor, but that we feel there is not much interesting to say. R2 can easily verify by checking the differentiation profiles in Supplementary Figure 2 – there is no broad-scale heterogeneity in population differentiation that would mirror an impact of heterogeneous recombination (for striking counterexamples in lake-stream stickleback see Roesti et al. 2012). The interesting exception is the signal in the contact zone addressed in our CCBD analysis. Apart from that, variable recombination does not influence our study in any relevant way. (The reason is given in the Berner & Roesti 2017 theory paper: there is no gene flow between Misty lake and stream stickleback outside the contact zone, see also our response 2). We cannot see how expanding the discussion of recombination would add quality to this paper.

Overstating the case for complete reproductive isolation – there are several points in the MS where the authors make some strong suggestions that reproductive isolation is near complete in the Misty system (i.e. LN181; LN244). This is an overinterpretation of the results. It is in fact possible to qualitatively estimate the extent of RI from hybrid zones using hybrid index but of course, the use of pool-seq data here means that is not possible. The authors have not actually measured reproductive isolation in this case – they are instead examining allele frequency differences between populations. I certainly agree with them that there is good evidence for strong divergent selection in this system and that this has shaped a strong cline in allele frequencies based on genome regions which might underlie morphological and ecological divergence. However, allele frequencies can be influenced by other processes too (i.e. linked selection, drift) and furthermore, allele frequency differences alone are poor estimators of effective migration rate (that is, reproductive isolation at the level of the genome – see Whitlock & McCauley 1999). A more appropriate interpretation of the pattern of increase in stream allele frequency is that there is stronger selection against the dispersal of lake alleles with the increase in distance from the tension zone. It is worth noting here that my earlier suggestion of a formal estimation of cline parameters for loci might also give some indication of how different genome regions differ in their dispersal from the cline centre, which in turn might lead to a more insightful analysis with regards to gene flow and introgression across this habitat transition.

10) R2 here raises several issues. The first is that we apparently do not measure RI, and that we overstate the completeness of RI. We disagree. Our initial visual analysis already made clear that beyond a narrow contact zone, allele frequencies at both fixed and random SNPs were stable, strongly suggesting that RI is near completion genome-wide. However, we agree with R2 that there was opportunity to address this issue more convincingly. A first important improvement is that we have re-defined the markers used for exploring gene flow. As described in our response 2 above, we now work with three panels of SNPs that should adequately tag strongly selected genome regions (the ‘selected’ SNPs) and regions plausibly evolving primarily by drift (our ‘neutral’ and ‘loDiff’ SNPs). Visually, these three SNP categories exhibit identical cline shapes (apart from the obvious difference that the maximum AFD differs – this follows from their ascertainment) (Fig. 2b and Supplementary Figure 4), and formal cline modeling offers no indication that these marker categories differ in cline center and width (Fig. 4, Supplementary Figure 6). Combined, this in our view offers compelling evidence that RI must be complete and homogeneous across the whole genome. We have taken care to improve the wording to make this insight clear (L208-211). Note also that in this passage, we give a precise definition of what we consider complete RI within the context of our study (L201-211), and that this RI refers to the lake and stream population as a whole, that is, outside the contact zone (L208-209).

Second, R2 emphasizes quantifying RI using the hybrid index. We fully agree that scrutinizing hybridization more directly using admixture estimates based on individual-level sequence data (i.e. haplotypes) would be interesting, and we now discuss this as an outlook for future follow-up work (L298-302). However, as we explain above, this appears primarily meaningful across the contact zone, as all our analyses indicate the absence of gene flow beyond that zone. Third, related to the previous aspect, R2 seems to imply that one cannot infer barriers to gene flow based on allele frequency data (hence population structure), referring to Whitlock & McCauley 1999 Heredity. We cannot follow. What that paper criticizes is transforming estimates of population structure

into estimates of migration rates (Nm). This requires plenty of theoretical assumptions that are usually violated in the real world (e.g. the absence of selection), so that this practice has rightly been discredited; our study of course makes no such attempt. Instead, we use our allele frequency data directly for inference based on population structure – fully in line with Whitlock & McCauley's 1999 view that 'studying the genetic structure of a population is essential to understanding its evolutionary properties'.

Fourth, R2 seems to express that allele frequencies are not only influenced by divergent selection, but also by linked selection and drift. Of course – this is what our study is all about: genome-wide linked selection causes strong RI between the lake and stream population (indicated by clines at neutral loci coinciding exactly with clines at the selected loci, see response 2), and this blocking of gene flow allows their separate evolution by drift, which we observe as divergence in allele frequencies. We cannot follow why R2 thinks that our analysis ignores these processes.

Finally, R2 proposes that cline modeling for different genome regions (i.e., under selection vs neutral) would inform to what extent gene flow (and hence RI) varies across the genome. This is a good point that we have addressed, see our response 2 above for details. In short, R2's 'different genome regions' are now represented by three categories of SNPs plausibly tagging selected versus two types of neutral regions. All these SNPs show the same cline parameter estimates (center location and width), confirming our conclusion that RI is genome-wide and essentially complete outside the narrow contact zone.

Minor comments

LN47-48: It is quite a strong statement to say that comparisons between ecologically divergent species largely ignores the eco-geography of speciation. On the contrary, there are many examples where this is an important aspect of the experimental design of these studies. Martin et al 2013 is a good example of this, where eco-geography is explicitly used to look at different levels of differentiation between species. Similarly, contact zones have long been recognised as areas of importance for understanding how RI translates to the genomic level. So, while I agree clinal analysis has not been well used in the early years of the genomic era, this is clearly being rectified now (as is seen by the references cited in LN51). I think this justification for the study needs to be reframed.

11) We have updated the text to be more precise (L44-48): of course, the majority of the population genomic studies consider different habitats/ecologies within some broad geographic context (mentioned on L40-41), including Martin et al. 2013 mentioned by R2 and much of our own previous genomic work (e.g. Roesti et al. 2012, 2014 Mol. Ecol., Haenel et al. 2019 Evol. Lett.). However, we here now explicitly refer to 'fine-scale eco-geography' (L45) and to 'a high geographic resolution' (L48), and with that imply 'a clinal focus' (L48-49). It should now be clear that we talk about a research framework quite distinct from the one typical in population genomics. Regarding genetic as opposed to spatial resolution, we are aware of only a single study having leveraged full-genome sequence data for clinal genomic investigation (mentioned on L57-58); all other references on L48 (formerly L51) are based on reduced-representation sequencing, which has well-known limitations (Lowry et al. 2017 Mol. Ecol. Res.). Hence, we feel it is fair to consider our study outstanding in its combination of fine spatial grain and full-genome resolution, a view generally shared by the reviewers: 'The study of a contact

zone at this level of resolution (both geographic and genomic) offers an unprecedented power for understanding speciation' (R3).

Fig 4 and the L1-S1 comparison – not really clear. I understand the method on rereading but the rationale is not obvious – probably needs better explaining

12) We understand R2 struggled with this part – this analysis is the most involved of all. To clarify, we have updated the description of this analysis in the main text (L253-260) and in particular in the Methods (L551-554 , L592-597). The latter update includes a more explicit statement regarding the statistical prediction and its interpretation. We hope that with these improvements, this analysis is easier to understand.

LN65: Paraptry – typo.

13) Thanks, corrected.

LN69: What does genetically compatible mean? Presumably no evidence of sterility or effect on viability. I think it is better to say no obvious genomic incompatibilities.

14) Ok, text updated as suggested (L63-64).

LN77: Given the context of this system and lake-stream stickleback in general as a model for parallel (or non-parallel) evolution, it's probably better to avoid the word parallel here. It confused me on first glance!

15) Ok, wording is changed and now unambiguous (L69)

LN82: Another example of a need to be careful with a word here. Effective could cause confusion with a term such as effective migration (which is actually very relevant in this case as clearly there is a high probability for migration of individuals, the question is whether their genes migrate too). I would suggest replacing this.

16) Ok, wording is improved (L80)

LN96: Name the sites for clarity. Are these distant stream or lake sites? Or between the lake and the stream (i.e. as in LN105).

17) Ok, good point. We now name the sites explicitly (L93-94).

LN111: For the sake of comparison, citing the F_{st} value would also be helpful. This is largely for others reading the study and to make it at least partially comparable to other genomic data (i.e. for example – the work of some of the authors!)

18) Ok, we now provide this specific differentiation value based on F_{st} (L108). As additional resources, we now present median differentiation as both AFD and F_{st} for ALL relevant pairwise sample combinations (Supplementary Figure 12). For the key L1-S7 sample comparison, we now additionally provide the distribution of AFD and F_{st} across all genome-wide SNPs (Supplementary Figure 3).

LN116: what determines a region? From Fig 3 it looks like there are only 3 SNPs which are actually fixed. I don't doubt the evidence for high levels of differentiation, but are the regions defined arbitrarily? Otherwise, why not use a method such as a HMM in order to identify regions of elevated AFD?

19) Good point, this passage was not fully clear. We here do two things: on the one hand, we provide a *qualitative* interpretation of genome-wide differentiation patterns by

characterizing ‘baseline’ differentiation (median AFD), and emphasizing that some genome regions stand out from this baseline. One can appropriately appreciate these aspects from visual analysis alone (Fig 3, Supplementary Figure 2). Second, we *formally operationalize* a panel of SNPs tagging genome regions under the strongest divergent selection in this system (given our full-genome resolution, this panel plausibly includes a fraction of causal polymorphisms *directly* under selection). This identification of ‘selected SNPs’ (formerly the ‘fixed SNPs’ panel) for further analysis involves objective and reproducible filtering steps (magnitude of differentiation; autosomal SNPs only; spacing to ensure independence). We have improved this latter part through substantial re-analysis, and while re-drafting, took great care to better separate the qualitative interpretation of patterns (L110-114) from our approach to define selected loci (L114-122, L477-488). To make the latter even clearer, we now present as Supplementary Figure 3 the genome-wide distribution of differentiation values and our cut-off delimiting high-differentiation SNPs. This graph should illustrate that our panel of ‘selected SNPs’ is chosen quite stringently; we doubt that a model-based approach – requiring its own assumptions – would lead to a more meaningful set of loci under selection. Note that we have applied (and internally validated) very similar approaches to delimiting SNPs representing genome regions under selection in previous work (Roesti et al. 2015 Nat. Commun.; Haenel et al. 2019 Evol. Lett.; Laurentino et al. 2020 Nat. Commun.). And of course, cutoff-based delimitation of loci under selection is common-place in the literature.

LN128: Without a proper example of what the authors mean by regions, it is very hard to actually assess whether this is the case – again, what is a differentiated region?

20) See our previous response; as now described in adequate detail in the main text, we apply an objective algorithm for identifying SNPs ‘representing putative genomic targets of strong divergent selection’ (L122). This algorithm is arguably stringent (AFD ≥ 0.97 threshold) and cautious (50 kb spacing between high-differentiation SNP clusters), and the resulting 50 independent loci we consider under strong divergent selection must just represent a small, conservative subsample of the total loci under selection in this system (Supplementary Figures 2 and 3). We see no problem with our conclusion in the passage indicated by R2, and no other reviewer struggled with this point.

LN151: Here again, the word effective can be confusing. Do the authors mean that the barriers work? I think it is important to be careful with the wording used to ensure the correct interpretation of the results.

21) Ok, wording changed (L162)

LN165: Inviability – however these are F1 crosses right? What about in backcrosses or F2s? Also inviability is only one component of incompatibilities – has hybrid sterility been checked?

22) Some of the experiments cited (including in Misty stickleback) were run over multiple generations and *did* include F2 hybrids and backcrosses. This is now specified (L176-178). We have also relaxed the wording to cover both viability and sterility.

LN173: I don’t think you’ve really shown reproductive isolation – you’ve demonstrated evidence of genetic differentiation which points towards a level of RI here. I think this needs to be softened.

23) We disagree with R2 – our inference of strong RI is uncontroversial (see our responses 2 and 10 above). To briefly recapitulate, how could genetic clines concordant among any marker category, and narrower than the distance this organism can disperse in a single day, be maintained in the absence of strong RI? We do not see any problem with this statement (L186).

LN180: I do not follow the logic here – this does not mean RI is complete. Selection against introgressed stream alleles means they cannot move beyond S4 in the contact zone.

24) Yes, exactly! If there is no introgression beyond S4, this means that the two parapatric populations at large (i.e., outside the contact zone) are completely reproductively isolated. Note that later in this sentence, we are fully precise about what we imply by complete RI (L210-211). For further details on this RI issue, see again our response 10.

LN215: Substantially stronger – but not really justified other than geographically – what is the height of this peak and the actual difference in AFD between the two comparisons?

25) A formal window-based operationalization of these high-differentiation regions based on the combination of the L1-L2 and L1-S1 comparisons is given in the Methods (L554-563). As this procedure is somewhat involved, we decided to keep the Results & Discussion passage referred to by R2 conceptual, without technical detail. However, we here still make clear that that these high-differentiation regions emerge relative to a reference comparison (L1-L2; L246-250), and we feel Fig. 3 makes really clear what we mean by relatively high differentiation in L1-S1 compared to L1-L2. But to address R2's issue, the quantitative detail requested (AFD difference between the two comparisons at HDW and non-HDW) is now added to the Methods (L559-560, 562).

LN220: Why not also look for regions

26) We do not understand this question. Our analysis focuses on regions (chromosome windows), see L557-561.

LN238: CCBD needs better explaining here.

27) Ok, we have rephrased (L277-281)

LN244-247: This is really overstated – it is not clear that reproductive isolation is complete here and you have not actually directly measured RI – thus you cannot state this.

28) Again, we disagree, see our response 10. Our analyses (visual and the new cline modeling) clearly allow us to draw strong conclusions about the completeness of RI. The conclusion in this passage is fully supported by evidence in our view.

LN279: "Were sufficient"

29) Thanks, updated (L326).

LN298-300: What about other inversions though? Is there any evidence of this from the current data? It is quite possible that other inversions segregate in lake-stream fish

30) Clearly, inversions do not play a relevant role in this study. Inversions are very easily detected even with sparse reduced-representation data and hence emerged prominently in early stickleback genomic work, which sometimes led to the biased view that inversions make a special contribution to adaptive divergence. However, the importance of inversions in this species complex vanishes when one increases marker resolution and hence the potential to capture signatures of adaptation at a finer physical scale. Our genome scans in Misty reveal absolutely no indication of divergence at inversions.

LN312-320: Hybridization between lake and stream fish could also drive the more extreme PC1, PC2 body shape differences compared to L2. Transgressive segregation among hybridising lineages can drive this – although it is difficult to test for.

31) Good comment. However, note that the phenotype values from the marsh site are not different from those from the first lake site (L1); what seems extreme are the values at site L2, which cannot be explained by transgressive segregation because the L2 sites should not be influenced by hybridization. Certainly, the reason for the relative noisy phenotype data is simply measurement error, which is usually large in such geometric morphometric analyses. Anyway, these data are shown just to provide a qualitative phenotype-genotype link (now stated explicitly on L423-424), so we prefer not to expand on this issue.

LN23-324: The flood event is an interesting addition to the paper – how often do such floods occur? Is it likely they are semi-regular events that would introduce migrants into the marsh environment? If so, I think it is worth emphasising the role of ecological/environmental cycles on passive migration – perhaps often overlooked in the literature with regards to dispersal between environments.

32) Based on ample field experience over decades by Andrew Hendry and his team, this flood was truly exceptional. But we have no hard data to quantify this further. Even precipitation records do not appear to predict water levels accurately. Hence we prefer not to expand this passage to avoid speculation.

LN338-339: Is this greater than expected by chance though? This could probably be simulated, given the estimates of the proportion of fish migrating into the marsh (i.e. Supp Mat Fig 5).

33) While the sudden decline in stream alleles in the marsh during the flood cannot be driven by selection, the slight rebound after one year could be due either to selection, to some immigration from the stream, or both. We mention these possibilities (L388-389), but given this ambiguity, simulations would be very difficult to conceive and unlikely to be conclusive.

LN432 & 435: Why 100Kb? Is this where LD decays to the background rate in stickleback?

34) Decay of strong LD typically occurs over somewhat shorter distance in stickleback, perhaps some 5-10 kb, but this depends on the local crossover rate. Our spacing threshold of 100 kb thus certainly ensured statistical independence of the SNPs, but was likely overly prudent. For this revision, we have carefully re-inspected the distribution of inter-marker spacing for SNPs showing extreme differentiation, which revealed that 50 kb is still a conservative threshold to avoid clusters of tightly linked SNPs. So we have

redone all analyses with this new threshold (which had no effect on any result). Note that we have used similar thresholds in previous work (see our response 4).

LN886-887: I may have missed this but what qualifies as strong lake-stream differentiation here? Is it the same as LN569-571? In which case, this should also be explained in the results & discussion.

35) Thanks for pointing out; indeed this detail was missing in the R&D section. We have added both to the main text (L376-377) and to the Fig. 7 legend.

Data sharing – why not share the allele frequency data too? And the genotype calls?

36) We have poolSeq data, hence no genotype calls. But we agree that sharing the raw nucleotide counts at all genome-wide positions in all samples may be a useful resource for future work; we will do.

Reviewer #3 (Remarks to the Author):

In their manuscript entitled “Clinal genomic analysis reveals strong reproductive isolation across a steep habitat transition in stickleback fish” Haenel and co-authors focused on a well-characterized stickleback population pair adapted to lake and stream habitats in Misty river, with the main goal of understanding (at the genomic level) how adaptive divergence results in strong reproductive isolation. The implementation of a whole-genome pool-seq approach to characterize multiple populations across an environmental transect from lake to stream habitats, complemented by the analysis of phenotypic traits, revealed steep clines of allele frequencies for multiple markers across the genome, with cline centres roughly coinciding with the habitat transition. This pattern, combined with individual-based simulations, was interpreted as evidence for strong polygenic adaptive divergence forming a genome-wide barrier to gene flow without physical isolation.

The study of a contact zone at this level of resolution (both geographic and genomic) offers an unprecedented power for understanding speciation at different levels (from genotype to phenotype). Thus, I think it can be of wide interest for researchers working on many different systems in the field. While I found this study powerful and highly interesting, I have a number of significant concerns that are important to consider.

i) Although I found the manuscript well-written, I think that the central question was not very clear. By the end of the introduction, the authors mention that this study aims to identifying reproductive barriers between lake and stream sticklebacks (in the Misty river system) at the genomic level. However, the impact of the different reproductive barrier types on gene flow was not assessed, and the analysis of SNP clines alone can hardly allow to identify which traits or components of isolation contribute as barriers to gene flow.

37) Good point, this indeed sounded like we are studying specific barriers. We have rephrased to make clear that our focus is on overall RI (L82-83), although we later do include a discussion of some underlying barriers certainly or potentially contributing to overall RI.

ii) I understand the constraints in terms of space but it would be important to have some

more background about the system. For instance, are we dealing with a primary or secondary contact? Another important aspect concerns stickleback migration. Do they migrate to breed in this lake/stream system? If they do, it would be important to know when was sampling conducted relative to breeding migration. Finally, some reproductive barriers in this system and their putative influence in the observed patterns should be described in more detail. In particular, the authors refer that sexual selection and habitat preference are unlikely to cause massive differentiation across small distances but they do not explain why. In general, I think that habitat choice is one way in which clines can be steep relative to dispersal, generating differentiation at many loci across the genome.

38) Demographic context: it is notoriously difficult to determine whether speciation occurred under continuous gene flow, or involved phases of physical isolation (secondary contact). We initially intended to incorporate in this paper demographic modeling based on indSeq RAD data from previous experiments. However, after months of cluster computing, we suspected that the results were unreliable, and checks with simulated data confirmed that in this system, selection biases the results from demographic software (which assumes neutral evolution). So we gave this up. However, there are two pieces of biogeographic evidence making it almost certain that this system represents primary divergence in the face of gene flow. We agree with R3 that these ideas may be relevant to the readers, hence have added to the MS (L70-78). There is additional evidence of primary divergence implicit in our paper: as we describe, there are no intrinsic reproductive barriers in this system (L176-178); but the evolution of precisely such barriers is generally considered the crucial consequence of evolution in isolation (i.e., secondary contact). Second, our simulation experiment supports the idea that genomically widespread selection, even when weak, will lead to RI despite initially high gene flow (L338-340).

Migration: We declare that we are studying a 'population pair residing in parapatry (that is, contiguously) in Misty Lake and its inlet stream', which in our view is fairly explicit. Moreover, migration in this system is highly implausible biogeographically (Fig. 1a; where should the lake and stream populations go, and why?), so we prefer not to expand on this point further.

Reproductive barriers: we have now improved the discussion of barriers likely to contribute to RI, see our response 5.

iii) My main criticism to this study is that despite the suggestion (abstract and introduction) that a formal cline analysis will be performed, I was surprised that this was not the case. I am not saying that plotting AFD across the transect is not at all informative. However, there is a body of theory and tools explicitly developed to extract rigorous information from the type of data the authors collected, such as: quantify how many loci present a significant cline and their distribution across the genome, quantify barrier strength, test differences in cline centres and widths, test for asymmetrical clines, etc. Cline analyses would also allow to explicitly test specific claims about polygenic adaptation (where stepped clines are expected), about allelic frequencies in S1, and about temporal instability of frequencies after environmental perturbation.

39) Similar feedback was given by R1 and R2, so we have performed a formal cline analysis as requested, see our response 2 above. As described, this new analysis addresses the key aspects mentioned by R3, and confirms that all our previous interpretations based on visual inference were correct.

iv) There are other methodological aspects would benefit from further improvements: the phenotypic data does not completely match the genetic sampling and is not much used in the analyses, which significantly weakens some inferences; it is not clear why the authors only used 500 random snps representing the non-fixed differences (even for these not much detail is given about the variability in clines and how are they distributed across the genome); it would be useful to know the uncertainty (error distribution) around the allele frequency estimates from pool-seq data; there is no information about recombination rate, despite the fine-scale maps available for stickleback. Finally, although I am fully aware of the problems related with FST, I am also not entirely sure that AFD (Berner, 2019) is free of biases. Would not be useful to compare the main results using AFD vs FST?

40) Phenotypic analysis: there have been several purely phenotypic studies in the Misty lake-stream stickleback system (cited, e.g. L67), so the purpose of the phenotypic part in the present study is just to forge a qualitative link between genomic and phenotypic data. We agree that this was not made clear enough in the paper, so we have updated (L423-424). We are also fully explicit that some of the phenotypic data are re-used from published work (L433-437), which explains why the sites do not fully match.

Random SNPs: Agreed, choosing 500 SNPs at random was perhaps suboptimal. As we explain in our response 2 above, we no longer work with a random panel of SNPs. Instead, we now use two panels of SNPs chosen carefully to tag regions under largely neutral evolution. Moreover, our cline modeling now describes variation in cline features among subsets of these SNPs in detail (Supplementary Figure 5).

Uncertainty in allele frequency estimates: the error introduced by the pooling of individual DNA cannot be known. Also, we did not infer genotypes but used raw allele frequency counts (now declared: L469-471), so there is no additional uncertainty from genotyping. What is certain from poolSeq theory, however, is that with sample size of c.56 per locality (L91) and an average read depth of c. 100x (L95), our study must estimate population allele frequencies with excellent precision (stated explicitly: L453-455). Just for comparison: the only other clinal study using whole-genome sequence data (Rafati et al. 2018 Mol. Ecol.) also used poolSeq, but sample size was just 10-20 individuals per site and read depth per pool was 10x. The quality of our allele frequency data is clearly outstanding.

Recombination rate: no, we do not talk a lot about recombination rate because we see no reason why this would be interesting. For details see our response 9 above.

AFD vs FST: we are not clear what R3 means by 'biases' – we are interested in how allele frequencies change along a spatial gradient, hence AFD is the ideal differentiation metric directly expressing the quantity of interest. Nevertheless, we have now redone some key analyses based on FST, presented as Supplementary Figures 3 and 12.

v) Given the effects of recombination (CCBD analysis), I would like to hear from the authors about the possibility of some of the AFD peaks resulting from background selection. We know that this system is relatively young and probably not many mutations occurred since divergence; it is known from previous studies that there is divergent selection in this system; and the cline centres seem to be located at the habitat transitions. However, I was wondering if some of the peaks (mainly in the central regions of chromosomes where recombination is lower) could have resulted from background

selection. Gene flow could have erased its signatures in the extremities of chromosomes close to the habitat transition where gene flow is higher but it might not be enough to counteract it in the centre of the chromosomes. This may not be strongly plausible but I would like to know the authors opinion about this alternative interpretation.

41) A discussion of background selection certainly has no place in this study. As R3 recognizes, this is a process requiring deleterious mutation, and the purging of these mutations causes only a very weak sweep because the deleterious alleles occur at low frequency. So a signature of background selection would require a time scale orders of magnitude longer than the age of our system. Moreover, background selection cannot drive a sharp peak because this would require that deleterious mutations occur deterministically at the same spot; background selection generally drives differentiation at a broader physical scale. Finally, the local erosion of such differentiation by gene flow is not plausible because there would be no force maintaining differentiation in the face of gene flow in the absence of divergent selection. Anyway, in postglacial stickleback systems, we can safely ignore background selection.

vi) Concerning the simulations, I think that the drift-alone model is not an appropriate null because it is known to be selection. The genome-wide effects of the barrier to gene flow can originate clinal patterns at sites that are unlinked to selected loci (see for example Westram et al. (2018) for a similar approach). The authors mention that their empirical results are compatible with polygenic selection they modeled but no alternative models (other than drift-alone) are tested and there was no formal comparison to the observed data. The core simulation results reported are focused on the model with unlinked loci, but the chromosome centre effect shows that recombination matters. The authors mention that they additionally simulated physical linkage among all loci along a single chromosome with recombination but it is not clear what is the LD between loci, how LD decays with distance and why a single chromosome. Since the authors likely have access to genetic maps and know the marker positions, I think it would be important that their “standard simulation models” incorporate recombination and to say more about the distribution of clines across the genome to support their claims about a genome-wide barrier.

42) We have a hard time following these comments. First, we not understand why R3 thinks that we use a drift-alone model; we declare unambiguously that we model exclusively a divergent selection scenario (‘We modeled polygenic divergent selection’, L313-314, 630-633), and Fig. 5 shows that we specify non-zero selection coefficients only.

Also, how could clinal patterns emerge ‘at sites that are unlinked to selected loci’? If RI is genome-wide, the entire genome is in LD, hence EVERY base position is linked to selected loci (although of course not physically).

No ‘alternative models’: what kind of alternative model does R3 have in mind? What evolutionary processes other than divergent selection and drift would matter in this system?

Yes, we avoid a formal comparison between our modeled and empirical data. Our model is an abstract representation of a system based on several assumptions; its purpose is in our view to reveal whether an observed pattern can be reproduced *qualitatively*. We feel a quantitative comparison would not be meaningful.

Chromosome center effect and recombination: we observe a signature of recombination modifying the selection-gene flow antagonism right at the transition zone

only, but not beyond (Fig. 2c), highlighting that recombination is not a relevant factor in this system (see also our response 9). Accordingly, we modeled a uniform distribution of crossover, which we now declare (L658). And we cannot describe how LD decays because LD is a function of gene flow and selection, hence specific to a given simulation. Also, we modeled only a single chromosome because this is the extreme opposite to free segregation – if one obtains similar patterns with free recombination versus a single chromosome with crossover, this indicates that recombination plays no role (and that modeling any higher number of chromosomes, or heterogeneous crossover rate along chromosomes, would not change the results). This is exactly what our model shows, and what the model of Flaxman et al. 2014 Mol. Ecol. indicates as well.

Distribution of clines across genome: see our response 2 – our new analyses (visual and cline modeling) now demonstrate unambiguously that loci across the *entire* genome show similar cline shapes, as expected under complete RI.

All in all, I think the system is quite interesting and can be very informative but I think the data should have been analysed in a more thoroughly manner to support the authors' main claims. Currently, I think the support for sentences such as: "This strong polygenic adaptive divergence must constitute a genome-wide barrier to gene flow because a steep cline in allele frequencies is observed across the entire genome..."; "Our simulations support our empirically-based conclusion that adaptive divergence from abundant standing genetic variation has produced strong reproductive isolation in the absence of physical barriers"; and "Collectively, our analyses indicate that reproductive isolation between the Misty Lake and the inlet stream population as a whole is nearly complete, allowing the evolution of the two populations largely unconstrained by gene flow." is rather weak. In fact, it is not clear to me why the authors conclude that reproductive isolation in this system is nearly complete. If we study populations far enough from the contact zone, there will likely be substantial differentiation even if selection is weak. How much differentiation also depends on time and population size. Once more, I think the cline analyses would be useful to measure the barrier strength.

43) Our new analyses (both cline modeling and visual) based on the selected, neutral and loDiff marker categories now make it absolutely clear that all conclusions highlighted above by R3 are fully supported by evidence. Also, we have improved the text to make this clearer. All this is detailed in our response 2.

Minor comments:

L22. "Dense" is vague and non-informative.

44) Good point, this specifier did not add much, is deleted.

L23. Same for "numerous".

45) Here we disagree. Of course we could put 'dozens' or 'hundreds', but such counts remain vague because they depend on analytical decisions. For instance, our analysis is certainly quite restrictive in identifying regions influenced by divergent selection, as we go for (nearly) fixed SNPs only. So we prefer to keep our wording; in the subsequent sentence, we say 'such polygenic adaptive divergence', so it should be clear at least qualitatively what 'numerous' means.

L44. Remove "genetic".

46) Ok, deleted.

L45. They can also harbour loci involved in incompatibilities.

47) If R3 thinks about intrinsic incompatibilities (BDMIs): no, not in a system as young as this one, and in which divergence almost certainly occurred in primary contact.

L60. Why only initiating?

48) Because even if ecologically-dependent RI is complete (= no more gene flow), speciation will typically remain reversible. Hence for complete speciation, sufficient intrinsic incompatibility is needed, which is a long-term process not necessarily requiring divergent selection.

L76-78. Can the authors present some numbers?

49) Ok, we now give F_{st} (0.12) as measured in previous work (L70). This is close to the genome-wide F_{st} value that we now present for our study too (0.14; L108)

L84. Replace “Mistly” by “Misty”.

50) Thanks, well spotted.

L92. Is “Results and Discussion” together as single section a valid format for Nature Communications?

51) Yes.

L97. I think the authors should indicate here the number of samples used in each pool (average and range).

52) We say that N per site approximately 56 in the preceding sentence. Full details are given in Supplementary Table 1.

L116. Are they fully independent? Does LD decays in less than 100kb in this system?

53) Addressed in our response 34 above.

L128. Not clear how the authors go from 34 to hundreds. Could not the target of selection be a single gene (or regulatory) per region?

54) We have redefined our ‘selected’ SNPs based on $AFD \geq 0.97$ (see response 2), which returned 162 SNPs near or at fixation. Imposing our strict spacing threshold now yields 50 *independent* regions. However, of course an AFD peak of 0.6, 0.7, or 0.8 in such a small-scale system must plausibly reflect a signature of selection too. That is, we are using only the tip of the selective iceberg for our analysis. That hundreds of genome regions are under divergent selection in this system is uncontroversial; just eyeballing Fig. 3, one can discern some 20+ regions on chromosome I alone that must plausibly be under selection.

L133. “action” instead of “presence”.

55) We stick to our wording because ‘action’ may suggest that selection is an agent, while it is just a process.

L138. Is not it actually a little bit shifted to the stream side? Is there any explanation for this shift?

56) Again well spotted. The first stream site, located exactly at the marsh-stream transition, has indeed lake-like allele frequencies. We later address this pattern explicitly and give an explanation: ‘this pattern led us to hypothesize that our stream site closest to the lake (S1) was overwhelmed by gene flow from the lake’ (L253-256). We re-considered carefully whether the pragmatic wording we use in the passage R3 refers to is adequate, and we feel it is.

L139-142. It is hard to see that it coincides. The shape of the clines is very different from the fixed SNPs. I think this needs to be properly tested before the authors conclude that it coincides.

57) Well, we do not have spatially well-resolved phenotypic data, but given the available resolution, the inference that the phenotypic transition occur at the habitat transition seems uncontroversial to us. And there is common sense too; given the evidence of genome-wide RI at the habitat transition, why should the phenotypic transition be elsewhere?

L144. Why only 500 SNPs? Please clarify if these can be located within the 34 highly differentiated genomic regions (even if they are not the 34 fixed SNPs).

58) We are surprised that R3 considers 500 SNPs sparse for quantifying a general pattern. Ten years back people would believe studies reporting genetic patterns based on 6 microsatellites (and the results were usually quite reproducible). But we agree with R3 that it was problematic to choose baseline SNPs by drawing at random. As described in response 2, we have redefined these SNP panels guided by AFD thresholds (we now work with 2x 500 SNPs). This makes it nearly impossible that they reside within a selected region.

L146. But the difference between their frequencies at both ends of the transect are much different.

59) Yes, and we explain twice why this is expected methodologically (L193-196, 529-532) – the SNPs were ascertained based on the frequencies in the terminal sites. To make sure, we have redone the whole cline modeling by excluding these terminal sites (presented as Supplementary Figure 7), which did not change any result.

L167-170. Not clear why they are unlikely to cause strong differentiation across small distances. Please explain. For example, why habitat preference cannot cause a strong barrier between lake-stream stickleback.

60) We have improved the discussion of these potential reproductive barriers, see our response 5.

L183. Not clear why “entire” is in italics. Same in L245 for “as a whole”

61) We have rephrased both passages.

L259-262. Can long distance dispersal occur in this system, even if at low frequencies?

62) Very hard to assess, both using genomic and field observation. If such dispersal were frequent, this would be seen in allele frequencies. However, we feel such dispersal is implausible: the factor maintaining strong RI in this system is polygenic divergent selection. Hence, no matter whether an individual dispersed only 150 m or 2 km from the

lake into the stream (or vice versa), it would almost certainly be eliminated by selection so rapidly that not even fragments of its genome managed to introgress.

L338-340. I see some changes in the distribution of allele frequencies but I think that is far from being offset, even if partially. Should not this be tested statistically? Additionally, I think that the use of tools like Admixture would be important to understand if there was actually some admixture to claim gene flow (L31) or simply immigration without gene flow.

63) We disagree with R3 on the first aspect. Please revisit Fig. 7 and compare the middle to the bottom panel: there is a quite striking decline in SNPs monomorphic for the lake allele (actually by 76%; we now mention this explicitly in the text, L387). So there is clearly something going on there. We agree with R3 that a formal test of whether the shift from the flood to the one-year-later allele distribution would be nice; we thought about implementing this. We realized, however, that this cannot be done by simply re-sampling the allele frequency distribution during the flood (i.e., the data in the middle panel of Fig. 7). The reason is that the allele frequencies one year later would, under the appropriate null model of drift alone (i.e., in the absence of gene flow and selection), be a function of local population size plus variance in reproductive success among individuals. Since we have no information on these points, we cannot generate a proper null distribution for this proposed test. Anyway, we feel the pattern is quite suggestive, and we have rendered the wording more cautious (L385-386), so our conclusion is certainly not going beyond the evidence. Also, we highlight that if individuals migrate, this does represent gene flow, no matter if they manage to reproduce or not. Hence our wording is uncontroversial in our eyes; no other reviewer struggled with this passage.

L395. Remove “by”

64) Done

L473. Why haploid individuals were simulated?

65) This does not matter; diploid vs haploid exclusively influences effective population size, which was here optimized to obtain a meaningful baseline magnitude of drift in the non-HDL.

L482. Why were all loci unlinked?

66) Because these loci are simulation replicates, hence need to be independent.

L542. Why not males and females?

67) Because it does not matter. Modeling hermaphrodite individuals is commonplace in theory.

L543. Why one offspring? Is this usual in sticklebacks?

68) Again, this modeling detail does not matter; an individual can mate multiple times, hence can leave multiple offspring to the next generation, like in a natural population.

L551. I found the number of replicates for these and the previous simulations to be relatively low. Can the authors somehow show information about the variance among replicates?

69) The adequate level of simulation replication depends on the stochasticity in the outcome. Replication in the order of what we use (N=20) is quite common in numerical theory (e.g. N=10-40, Flaxman et al. 2013 Evolution; N=10, Berner & Thibert-Plante 2016 JEB). Also, we considered presenting variance among replicate runs, but this appeared impractical given the vast parameter space we explore, and we see limited benefit in plotting individual simulation results for just a few selected parameter combinations. However, the lines in Fig. 6 are very smooth and well-behaved, which implies that the noise around the means cannot be large.

L559. How physical linkage was modeled?

70) We specify that all loci are sitting on a single chromosome with recombination, which necessarily implies physical linkage. We have improved the phrasing (L657-658).

L848. I am not sure we can call it strongly differentiated mainly when compared with differentiation between L1 and other S sites.

71) We are not sure we understand this comment. We clearly state in this legend that differentiation is strong in the L1-S1 comparison relative to the L1-L2 comparison, *not* L1 to other S sites. Given genome-wide baseline differentiation is otherwise very similar between these two comparisons, our interpretation seems uncontroversial to us.

L886. How strong?

72) Good point, this was not clear enough. We have improved, see our response 35.

Figure 5b. It is not clear which demes are fixed for the stream allele.

73) Good point (we suspect that R3 refers to the rightmost panel). We have added a note specifying that *all* stream sites are fixed.

Reviewer #4 (Remarks to the Author):

This study combines genomic data, morphological analyses, and simulations to demonstrate a strong polygenic basis of reproductive isolation between adjacent lake, marsh and stream populations. Specifically, this study used clinal analyses of phenotype and genetic variation along an habitat transition, using both morphometric analyses and pooled whole genome resequencing data. These empirical analyses are supported by spatially explicit genetic simulations of adjacent demes with different rates of gene flow and selection. Additionally, a natural event (flood) that occurred during the study provides support for the authors' conclusions. Overall I enjoyed reading this paper, and think that with a few revisions, it will be a useful contribution to the literature. Below are some suggestions.

General comments:

- Pooled WGS data was used, presumably to enable whole genome sequencing for many individuals, but pooled data does limit inferences in some ways and these should be discussed. In particular, I am thinking of the difficulty of identifying admixed or hybrid individuals from pooled data, and how that might limit understanding of this system, specifically as relates to the frequency and fitness of hybrids.

74) We feel one here needs to consider very carefully what type of evidence would critically require individual-level sequence data beyond the poolSeq data used. Our revised visual analyses and the new cline modeling now provide unambiguous evidence of quasi complete RI (see our response 2) – based exclusively on allele frequency data. That said, we agree that we have trouble quantifying the level of genetic mixing and the depth of hybridization within the narrow contact zone. However, even here specific analyses we perform give us an indication: both the small peaks in the L1-S1 comparison and the CCBD-peak in the contact zone are clear indications of hybridization (not just dispersal). We thus feel that by exhausting poolSeq data fully, one can get surprisingly far. However, we concur with R4 that a statement about indSeq data and how this would allow inferring the depth of genomic mixing was missing. We have added two passages making this explicit (L298-302, 396-399), and another statement is on L379-380.

- Claims of adaptive divergence require strong evidence that there are indeed fitness consequences to not having the putatively adaptive phenotype. As far as I can tell, there are no direct observations of fitness (of hybrid or translocated individuals) in this particular system. European examples of transplant/hybrid fitness are cited, but it is unclear how applicable they are to this system, which differs from other stickleback systems in several ways (e.g., in the polygenic arrangement of differentiation and absence of large inversions associated with ecotypes). Is there evidence beyond the coincidence of phenotypic, ecological, and genetic clines in this system for the increased fitness of individuals with putatively adaptive traits? Without direct evidence of selection, inferring a tension zone is potentially difficult, and a more cautious presentation of this inference might be warranted.

75) We are surprised by this skepticism toward adaptive divergence – can R4 propose a process other than divergent selection (i.e., a fitness tradeoff between the habitats) that would allow the fixation for alternative alleles in numerous genome regions, and massive differentiation across the *whole* genome (see our response 2), across <150m in a highly mobile vertebrate residing in a watershed without physical barriers? Agreed, there is no solid transplant evidence for fitness tradeoffs for this specific system, but there is overwhelming comparative evidence from habitat-phenotype-marker genotype associations across this and many other lake-stream systems from the same study region, for example:

Hendry et al. 2002, *Evolution*: ‘Adaptive divergence and the balance between selection and gene flow: lake and stream stickleback in the Misty system’

Hendry & Taylor 2004, *Evolution*: ‘How much of the variation in adaptive divergence can be explained by gene flow? An evaluation using lake-stream stickleback pairs’

Berner et al. 2008, *J. Evol. Biol.*: ‘Natural selection drives patterns of lake–stream divergence in stickleback foraging morphology’

Berner et al. 2009, *Evolution*: ‘Variable progress toward ecological speciation in parapatry: stickleback across eight lake-stream transitions’

In short, that there is adaptive divergence across lake-stream transitions in stickleback is established, and relevant literature is cited (e.g. L67, 74). Moreover, R4 states that the applicability of direct experimental evidence on divergent adaptation from a European lake-stream system is unclear. We disagree: first, polygenic divergence has been inferred in all lake-stream systems studied so far (stated: L128-129), and chromosomal rearrangements do not play a particular role in divergence (see response

30). Second, we describe explicitly that the Misty system is much more divergent both phenotypically and in genetic markers than the European system, hence adaptive divergence is plausibly even stronger in Misty stickleback (L168-171). Overall, our adaptive interpretations are certainly well supported by evidence.

AFD, allele frequency difference, is used as the primary measure of genetic divergence between lake and stream fish. While I think this is a valid choice, I would like to see some explanation of why AFD is used instead of F_{ST} , which is more common (yes, I know there's a whole paper about this and it's cited in this manuscript, but a brief justification would be helpful in this manuscript). I am also curious about whether results would look different if F_{ST} was used instead of AFD - does F_{ST} miss some of the differentiation, or result in different inferences? Is there a methodological point to be made here too for researchers working in other low-divergence systems or incipient species about AFD vs F_{ST} ?

76) Good point. We have redone several analyses using F_{ST} , which did not influence any conclusion (key results with F_{ST} are presented in the main text and Supplementary information, see response 18). An interesting detail, however, is that due to the low sensitivity of F_{ST} when differentiation is subtle, the informative peaks in the L1-S1 comparison may have been overlooked more easily when using F_{ST} instead of AFD. We feel this is a relevant methodological point, now mentioned in the main text (L250-253).

Specific items:

- Units of coverage are unclear (Line 403) - is it a mean of 103 reads per locus for pools of ~50 individuals, or 103 reads per locus per individual? I suspect it is the former, but these are very different levels of coverage, so clarification is needed here.

77) We have improved the wording (L452). It is now clear that individuals were pooled to libraries, and that the 103x mean coverage refers to these libraries.

-HDW and HDL are not defined explicitly at first use (Line 464, 474), although it is possible to infer meaning from the surrounding text. Please add definitions the first time you use these acronyms to make this clearer.

78) Ok, a definition is now given (L558).

Minor comments/typos:

Line 65 - "Parapatry" where it should be "parapatry"

Line 84 - "Mistly" where it should be "Misty"

Line 264-266 - awkward phrasing, sentence fragment

79) All minor comments fixed

Reviewers' Comments:

Reviewer #2:

Remarks to the Author:

This is my second time reviewing this manuscript (I was reviewer 2 previously). I think the authors have generally done a good job addressing my comments and those of the other reviewers. I think the incorporation of a formal cline analysis has helped bolster their conclusions of high and pervasive RI across the genome in this system. For the record I didn't think their previous approach had ignored the roles of linked selection or drift, just that cline analysis would help emphasise this and I think it does.

I am still a bit unsure about the use of complete RI in this study. I think this comes down to semantics and some debate as to what we mean when discuss RI in general. Clearly there is some level of gene flow across the transition zone, as the results of L1 vs S1 in Fig 3 demonstrates and the authors recognise in their response. However, they also show that gene flow attenuates with distance away from the transition zone – is this good evidence of RI? It could just as easily be low dispersal from the transition zone. As the authors state, the geographical structure of the system plays a role in mediating dispersal (LN267) and thus there is potentially a form of (micro)geographical separation here. Does that represent complete RI between sample sites at a good distance beyond the transition zone? I don't think so – they would still have no problem hybridizing if brought into contact. The solution here is not to focus on the completeness of RI then but its strength and effectiveness in limiting gene flow across the transition zone.

LN75: Doesn't make much sense to me why variation in progress along the continuum is evidence or an argument for the feasibility of primary divergence. One could quite easily argue the same in reverse – that this is a continuum of secondary contact with different levels of gene flow.

LN124-125: But the SNPs themselves tag regions and are not necessarily the targets of selection. If you take the 50Kb regions identified, what is the average number of genes within them? Is it different to the genome background or randomly sampled 50Kb regions across the genome? That would provide stronger evidence for the primacy of regulatory evolution in my view.

LN178: Does this include tests of hybrid fertility?

LN265: This needs qualifying – 'potentially harbors loci'

LN277-281: Is this really the main reason? I think a citation is necessary here, my understanding is selection is more efficient at removing deleterious sites and linked diversity, thus reducing within population diversity and maximising between population differentiation. And that this happens irrespective of the size of an introgressed region.

LN343-344: The literature is not limited to only to adaptive alleles linked by inversions – there are many cases where divergent selection maintains multiple divergent genome regions without physical linkage. The sudden mention of inversions here feels very out of place. I don't really feel like inversions need to be mentioned at all here.

LN383: Estimate of proportion of fish replaced – this is something that can actually be estimated with a simple ABC model to produce a parameter estimate with confidence intervals, rather than just a qualitative visual inspection. Csillery et al 2008 TREE should be a good introduction on how to do this. It's not essential but I think it would be a nice addition.

Supp mat: "Even if reproductive isolation is complete, genome regions exhibiting greater population differentiation by chance (stronger drift) or due to divergent selection will produce greater cline width estimates." – perhaps I'm misunderstanding here but doesn't Supp Figure 6 show the opposite of this? That cline width measurements are lower for regions under selection or drift?

I'm sure it was an oversight but there were no figure legends in the manuscript.

Reviewer #3:

Remarks to the Author:

General comments

I consider that the revised manuscript "Clinal genomic analysis reveals strong reproductive isolation across a steep habitat transition in stickleback fish" by Haenel and co-authors has substantially improved when compared to the previous version. Namely, the implementation of a formal cline analysis contributed to strengthen the support for some of the authors' claims. The clarifications of the study goals and the discussion on about the possible reproductive barriers on this system further contributed to increase the manuscript clarity and interest, respectively. However, I think that there are still some issues that need further attention.

First, the authors mention (several times) that the location of the main shift in allele frequencies and phenotypes coincide with the habitat transition. I touched on this issue in the previous round but I did not find the arguments provided by the authors very convincing (authors answer 57: "Well, we do not have spatially well-resolved phenotypic data, but given the available resolution, the inference that the phenotypic transition occur at the habitat transition seems uncontroversial to us. And there is common sense too; given the evidence of genome-wide RI at the habitat transition, why should the phenotypic transition be elsewhere?").

Please correct me if I am wrong, but is not the major habitat transition just downstream of S1 (i.e. between stream and marsh/lake)? If yes, the genome wide RI does not really coincide with the habitat transition. Or do the authors consider the main habitat transition between S1 and S2, even if these two are stream habitats?

Furthermore, under the available resolution is actually difficult to objectively say where is the transition in term of number of gill rakers and pelvic spine length, but if I had to define visually a cline centre, I would perhaps put it at the left of the habitat transition (I may be wrong and the authors may have a different opinion but this just illustrates the difficulties of not being possible to use cline-fitting for the phenotypic data). Additionally, I think there are reasons why the phenotypic transition could differ from the habitat transition, such as directional dominance or reproductive isolation caused by intrinsic effects not related with those phenotypes (although this last hypothesis is perhaps less likely in this system based on the authors description), among others. Finally, the authors did not formally test if the cline centres differ across different classes of markers. Could not this be implemented? Based on all these arguments, I think the authors should at least be much more cautionary (and precise) when referring to coincidence among the main genetic, habitat and phenotypic transitions.

Second, I am not an expert on cline analysis but I it is not clear for me if the authors first used cline-fitting to test if a given SNP has a clinal distribution or not. Actually, I am not sure if the package the authors used for cline-fitting allows to compare the likelihood between a clinal and non-clinal model (for instance using AIC)? However, this would allow to quantify the number or proportion of clinal SNPs across the three different classes of SNPs defined by the authors. Also, if a SNP is non clinal, I am not sure what the estimates of cline centers and widths mean.

Furthermore, I found the suggestion of a cline-fitting artefact creating wider clines for median and loDiff SNPS supported by simulations an interesting possibility. This may well be the case. However, this is a bit challenging as that is also the expected pattern from residual gene flow (incomplete reproductive isolation). The authors fitted tailed clines, but I think this can be problematic for loci without fixed differences because it is hard to distinguish the tail from the end frequency. Thus, given the relatively small number of sites, would not make sense to use a model without tails to confirm that this is really an artefact?

The third issue is relative to one set of simulations implemented by the authors (L425-428). I brought this issue in the previous round (point vi) but I do not think I was clear enough. What I meant is that for the specific simulations the authors describe in L841-871, the authors basically let the populations evolve under drift and compare the empirical ΔAFD distribution with the distribution of values obtained with the simulations. However, I think that a model with drift alone is not the best null model. The authors mention in their answer (42) "If RI is genome-wide, the entire genome is in LD, hence EVERY base position is linked to selected loci (although of course not physically)..." and in their answer (62) "Hence, no matter whether an individual dispersed only

150 m or 2 km from the lake into the stream (or vice versa), it would almost certainly be eliminated by selection so rapidly that not even fragments of its genome managed to introgress." Finally, the authors wrote in L.422-425: "This led us to hypothesize that our stream site closest to the lake (S1) was overwhelmed by gene flow from the lake, with appreciable genomic differentiation from the lake maintained in genome regions under the strongest divergent selection only. If true, these specific regions should, compared to randomly chosen genome regions, exhibit exceptionally strong differentiation between the lake and the stream population in general, including between the two samples from the endpoints of our geographic gradient (L1, S7)." Given this last statement and that there is genome-wide selection (two previous statements), I think that if the authors want to test whether those regions exhibit exceptionally strong differentiation compared with the rest of the genome, they should take into account that there is genome-wide selection when they let the populations evolve in their simulations. Any marker in the genome is under the influence of genome-wide selection and not just drift. Perhaps the conclusion does not change but I think that in this particular case, the influence of the genome-wide barrier should not be ignored.

Fourth, the authors mentioned in several answers that recombination is irrelevant in this particular study because reproductive isolation is complete (e.g. "Combined, this in our view offers compelling evidence that RI must be complete and homogeneous across the whole genome", or essentially complete outside the narrow contact zone. However, either RI is complete or not. If it is incomplete in the contact zone. Then, it is incomplete and there is potential for alleles to recombine away from selected loci and introgress. Unfortunately, pool-seq data is not very informative about admixture. Even when we see clines of allele frequencies with pool-seq, we cannot distinguish, for instance, whether allele frequencies in a given point of the transect is 0.7 because all individuals have approximately 0.7 and 0.3 of ancestry from each source population, or because 70 individuals in the pool are homozygote for one allele and 30 individuals are homozygote for the other. Assuming that there is antagonism between selection and gene flow and that gene flow in contact zone is heterogeneous across the genome, as the authors defend, then I think that linked selection and recombination should be important, even if in a relatively narrow zone of the transect. The authors mention that there are genomic regions that resist gene flow (between L1 and S1). Thus, it would be important to discern if this is just due to selection or actually a combination of the both the effect of divergent selection and low recombination. I think the CCBBD results somehow suggest that recombination plays a role in differentiation at least near the habitat transition. Probably this cannot be tested here but it is possible be that this antagonism between selection and effective migration (that also depends on recombination rate across the genome) is important to avoid fusion around the habitat transition and that those highly differentiated regions can be instrumental to achieve a more genome-wide differentiation far from the center. Of course is just a hypothesis but I hope I presented valuable arguments why recombination may matter in this system.

Other comments

L100. I do not fully agree that variable progress across the speciation continuum necessarily supports primary divergence (L100). I am not saying that this system does not represent primary divergence but variable progress across the speciation continuum is also possible across multiple replicates of a secondary contact between two populations/species. This may happen if incompatibilities are polymorphic, where each replicate can follow its own evolutionary trajectory.

L273. Steep clines are a function of dispersal distance and do not necessarily translate into near complete reproductive isolation. Thus, inference of strong reproductive isolation should take into account dispersal. Is dispersal distance known for this system?

L346-348. Not all clines go to the bottom at S1, as expected given the patterns observed in Figure 3.

L391-397. I do not follow why this asymmetry supports that there is reproductive isolation a few hundred meters above the habitat transition.

L461-464. Is not also because there is actually low recombination and consequently less effective migration on those regions, even if foreign chromosome bits are not maladaptive?

L510. Perhaps it would be useful to know the generation time of this species to know the time frame we are dealing here.

L537-580. I think it is good that the authors highlight the importance of polygenic selection in this system, as a contrast to the role of inversions in other systems, because it contributes to a more

balanced view of the genomic architecture of reproductive isolation. However, the documented perturbation in the system could suggest that this polygenic differentiation is possibly reversible whereas this may be less likely with chromosomal rearrangements, where incompatibilities can accumulate despite of the gene flow in the rest of the genome (Navarro and Barton, 2003).

L679-682. Were no mapping quality filters applied? (for instance, to remove reads that map to multiple genomic regions)

L700. After reading the reason why the authors applied the extreme (0.25) MAF cut-off (answer 7), I wonder if this cannot contribute to bias towards a pattern of stronger whole-genome reproductive isolation. I know that a MAF of 0.25 does not mean that SNPs will be highly differentiated between habitats but could there be an effect of MAF on differentiation and clinal patterns mainly for the non-selected SNP set?

Other issues raised in the previous round that I would like to clarify (after ' > ')

ii) Another important aspect concerns stickleback migration. Do they migrate to breed in this lake/stream system? If they do, it would be important to know when was sampling conducted relative to breeding migration.

38. Migration: We declare that we are studying a 'population pair residing in parapatry (that is, contiguously) in Misty Lake and its inlet stream', which in our view is fairly explicit. Moreover, migration in this system is highly implausible biogeographically (Fig. 1a; where should the lake and stream populations go, and why?), so we prefer not to expand on this point further.

>Perhaps I was not clear but my question was if fish could migrate between lake and stream to breed in a different place from where they spend most of their life; if yes, perhaps it would be important to know when was sampling conducted relative to the time of the breeding migration.

iii) Finally, although I am fully aware of the problems related with FST, I am also not entirely sure that AFD (Berner, 2019) is free of biases. Would not be useful to compare the main results using AFD vs FST?

40. AFD vs FST: we are not clear what R3 means by 'biases' – we are interested in how allele frequencies change along a spatial gradient, hence AFD is the ideal differentiation metric directly expressing the quantity of interest. Nevertheless, we have now redone some key analyses based on FST, presented as Supplementary Figures 3 and 12.

>I meant that Berner (2019) also described some bias for AFD estimates, although it may not apply to this specific case, given the high sample sizes. Anyway, I think it is good that the authors now use both AFD and FST, so that the main results do not depend on the method used to estimate differentiation.

iv) Given the effects of recombination (CCBD analysis), I would like to hear from the authors about the possibility of some of the AFD peaks resulting from background selection.

41) A discussion of background selection certainly has no place in this study...

>I thought could be beneficial to the reader that the authors just briefly explain why differentiations peaks they found are not likely generated by background selection but I am fine with the editor decision about this.

L60. Why only initiating?

48) Because even if ecologically-dependent RI is complete (= no more gene flow), speciation will typically remain reversible. Hence for complete speciation, sufficient intrinsic incompatibility is needed, which is a long-term process not necessarily requiring divergent selection.

>I think that a definition of speciation completeness that depends on irreversibility is not very useful because we will not be able to test it often. It also suggests that complete speciation is only possible if there are many intrinsic incompatibilities. However, this is just my opinion. I respect the authors' definition as soon as they are clear about it.

L97. I think the authors should indicate here the number of samples used in each pool (average and range).

52) We say that N per site approximately 56 in the preceding sentence. Full details are given in Supplementary Table 1.

> I think I was not clear but I was wondering if average and range would not be more informative. Anyway, this is a minor detail as N is actually similar among sampling locations. Perhaps the

authors should call Supplementary Table 1?

L128. Not clear how the authors go from 34 to hundreds. Could not the target of selection be a single gene (or regulatory) per region?

... That hundreds of genome regions are under divergent selection in this system is uncontroversial; just eyeballing Fig. 3, one can discern some 20+ regions on chromosome I alone that must plausibly be under selection.

> I am not really saying that there are not hundreds of regions under selection. I actually do not know. However, I think there should be a more rigorous way of defining and quantifying them than just eyeballing.

L482. Why were all loci unlinked?

66) Because these loci are simulation replicates, hence need to be independent.

> My question was not clear enough. I just wanted to ask why the authors did not modelled LD between markers, but it is now clear how this was done in the additional simulations with loci distributed across a single chromosome.

Responses to reviewer comments NCOMMS-20-22940A

We greatly appreciate the second round of evaluation of our manuscript NCOMMS-20-22940A by two of the initial reviewers (R2 and R3), which allowed us to further increase the quality of the paper. The comments identified many valuable issues deserving clarification; we have addressed these by numerous improvements to the text. In this letter, we describe what updates we made to the manuscript, and provide an explanation in cases where we disagree with the peers. The reviewer comments are given in complete and original wording in black; our responses are printed blue. R3 in some comments cites passages from the previous round of evaluation, printed gray.

Reviewer #2 (Remarks to the Author):

This is my second time reviewing this manuscript (I was reviewer 2 previously). I think the authors have generally done a good job addressing my comments and those of the other reviewers. I think the incorporation of a formal cline analysis has helped bolster their conclusions of high and pervasive RI across the genome in this system. For the record I didn't think their previous approach had ignored the roles of linked selection or drift, just that cline analysis would help emphasise this and I think it does.

I am still a bit unsure about the use of complete RI in this study. I think this comes down to semantics and some debate as to what we mean when discuss RI in general. Clearly there is some level of gene flow across the transition zone, as the results of L1 vs S1 in Fig 3 demonstrates and the authors recognise in their response. However, they also show that gene flow attenuates with distance away from the transition zone – is this good evidence of RI? It could just as easily be low dispersal from the transition zone. As the authors state, the geographical structure of the system plays a role in mediating dispersal (LN267) and thus there is potentially a form of (micro)geographical separation here. Does that represent complete RI between sample sites at a good distance beyond the transition zone? I don't think so – they would still have no problem hybridizing if brought into contact. The solution here is not to focus on the completeness of RI then but its strength and effectiveness in limiting gene flow across the transition zone.

1) A large body of research in lake-stream stickleback indicates that there is strong divergent selection between these habitats (L64-67). Moreover, some of our observed genomic signatures indicate hybridization within the narrow contact zone, hence reproductive isolation across the Misty Lake-stream transitions cannot be mediated by dispersal alone (i.e., some form of habitat preference), but involves a direct role of divergent selection. Nevertheless, we acknowledge in the paper that habitat preference (hence dispersal) likely contributes to reproductive isolation in this system (L182-184), and do not think that this is in tension to any of the conclusions in our study. That reproductive isolation beyond a few hundred meters from the contact zone is very strong is well-supported by our marker clines. How this isolation arises is less clear, we agree, but we feel that we are open-minded about the underlying mechanisms (L171-184). That reproductive isolation must somehow be linked to ecology also emerges from the observation that lake-stream transitions very often lead to phenotypic and genetic discontinuities in stickleback. We now mention this on on L72-75.

That said, we agree with R2 that our usage of the expression 'complete isolation' had the potential to create confusion; R3 similarly proposes to avoid this expression. We have now followed these suggestions and replaced 'complete isolation' by less categorical wording based on the strength of isolation. For details please see the first paragraph in our response 14 below.

LN75: Doesn't make much sense to me why variation in progress along the continuum is evidence or an argument for the feasibility of primary divergence. One could quite easily argue the same in reverse – that this is a continuum of secondary contact with different levels of gene flow.

2) This is a valuable comment, appreciated. What we here referred to is an argument originally raised by Jiggins & Mallet 2000 TREE as evidence of the feasibility of parapatric speciation. But we agree with R2 that by itself, variable progress along the speciation continuum is not a strong argument. We think that the collective strength of the evidence for primary divergence in the face of gene flow in lake-stream stickleback comes from 1) the consistent association of phenotypic and genetic discontinuities with an ecotone across numerous, evolutionarily independent watersheds (implying divergent selection as the main cause of reproductive isolation); 2) the absence of any intrinsic reproductive isolation in these young systems (again implying selection as a key driver of reproductive isolation); 3) the temporal stability of lake-stream differentiation where this has been investigated (not easily explained under secondary contact in the absence of intrinsic isolation); and 4) that parapatric lake-stream stickleback pairs are ubiquitous, hence evolve very easily. The latter is not compatible with secondary contact, as hydro-geographically, the initial evolution of physically isolated lake and stream fish in numerous watersheds is implausible. We have now revised this passage to move away from the Jiggins & Mallet argument, and to better expose the evidence that in our view is really compelling (L72-79). To further substantiate the above argument 4), we have here incorporated a new reference (Caldera & Bolnick 2008 Evol. Ecol. Res.) indicating spatio-temporally sequential, repeated lake-stream stickleback differentiation within a single watershed – an ecogeographic context in which invoking secondary contact is completely implausible. In short, numerous lines of evidence make it uncontroversial in our view that lake-stream divergence in stickleback, including in the Misty system, must reflect primary divergence. We hope that the revised passage now conveys this message convincingly.

LN124-125: But the SNPs themselves tag regions and are not necessarily the targets of selection. If you take the 50Kb regions identified, what is the average number of genes within them? Is it different to the genome background or randomly sampled 50Kb regions across the genome? That would provide stronger evidence for the primacy of regulatory evolution in my view.

3) Keep in mind that we have whole-genome physical resolution, and very precise allele frequency estimates thanks to large numbers of individuals per pool (>50; L94) and outstanding read depth (c. 100x; L98). It thus appears reasonable to assume that a good fraction of our 'selected SNPs' really *are* the direct targets of divergent selection. Hence, if we observe that just a single SNP out of 50 maps to a coding gene sequence, this must offer fairly strong evidence that coding variation is not very important in this system. (Note that we repeated this analysis to make sure, and found that we had previously overlooked a single SNP mapping to a coding gene sequence, now updated,

L128; however, this does not change our conclusion at all.) Nevertheless, we feel that we use rather cautious wording in this passage, L120-130. Moreover, although we examined only a single SNP from each of the 50 high-differentiation genome regions, this still represents c. one third of all genome-wide SNPs identified as high-differentiation markers (N = 162; L120). So the reduction to the selected SNP panel by subsampling the full panel is highly unlikely to cause major bias. Nevertheless, we have now expanded our analysis of whether or not SNPs lie in a coding gene sequence to the full set of 162 high-differentiation SNPs, which produced a very similar result supporting the same conclusion (now mentioned in the text: L130-132).

Regarding R2's suggestion to assess the number of genes around our selected SNPs – we cannot follow. Why would this number be informative on whether the target of selection is coding or regulatory – what is the prediction? Even if there were plenty of genes near a high-differentiation SNP, the target of selection could be a small regulatory (non-coding) element. In short, we do not understand why gene density should inform on the nature (coding vs. non-coding) of the polymorphisms. We feel our strategy to focus on the precise genomic position of our actual candidate polymorphisms is more conclusive.

LN178: Does this include tests of hybrid fertility?

4) Yet, it does. There is no evidence of *any* kind of postzygotic problems in young lake-stream stickleback systems, hence the overarching term 'intrinsic inferiority' is appropriate.

LN265: This needs qualifying – 'potentially harbors loci'

5) Thanks for this suggestion (i.e., to use 'potentially' to render the wording more tentative). We have carefully considered this specifier but think that it would generate a conflict with the evidence presented. The reason is that these differentiation peaks in the L1-S1 sample comparison *do* resist homogenization by gene flow, and our analysis using the L1-S7 comparison *does* demonstrate that these regions are under selection. We of course recognize the noisy nature of such data and cannot guarantee that all these high-differentiation regions are under selection, but if we used 'potentially', this would logically include the possibility that *none* of these focal loci are under selection, which is clearly refuted by our statistical analysis. We thus prefer not to add 'potentially', as this may confuse.

LN277-281: Is this really the main reason? I think a citation is necessary here, my understanding is selection is more efficient at removing deleterious sites and linked diversity, thus reducing within population diversity and maximising between population differentiation. And that this happens irrespective of the size of an introgressed region.

6) Yes, this is really the reason. There is a full-length theory paper scrutinizing exactly this process (Berner & Roesti 2017, Mol. Ecol.). This reference is given in the preceding sentence (L286, reference 54). As to the process described by R2's, what is missing is that under polygenic divergence, a larger introgressed region will on average hold more locally maladaptive alleles, hence segment length (and the distribution of recombination) matters.

LN343-344: The literature is not limited to only to adaptive alleles linked by inversions – there are many cases where divergent selection maintains multiple divergent genome

regions without physical linkage. The sudden mention of inversions here feels very out of place. I don't really feel like inversions need to be mentioned at all here.

7) Agreed, divergent selection generates gametic phase disequilibrium between populations, but we feel this is well known and not the crucial point here. This passage refers specifically to a branch of influential theory emphasizing the importance of true physical linkage among selected loci, as generated by inversions, for reproductive isolation. We strongly feel that this theory is being taken too literally, and that highlighting that our stickleback system is inconsistent with predictions by this theory helps the research field get a more balanced perspective. It seems we are not alone with our view; R3 expresses below that *'it is good that the authors highlight the importance of polygenic selection in this system, as a contrast to the role of inversions in other systems, because it contributes to a more balanced view of the genomic architecture of reproductive isolation'* (see the R3 comment that we answer in our point 21 below). We can understand that R2 find this passage somewhat out of context, but many readers will likely find it quite insightful, so we hope R2 understands that we prefer to keep this part as is.

LN383: Estimate of proportion of fish replaced – this is something that can actually be estimated with a simple ABC model to produce a parameter estimate with confidence intervals, rather than just a qualitative visual inspection. Csillery et al 2008 TREE should be a good introduction on how to do this. It's not essential but I think it would be a nice addition.

8) Excellent idea, appreciated – we had not used ABC previously and did not recognize its utility in this context. Having digested the recommended and other literature on the subject, we used ABC as suggested by R2. Specifically, we estimated the immigration proportion and its spread, and the result is in excellent agreement with our visual inference (method and results presented as Supplementary Fig. 11b). As suspected by R2, this new analysis does not change anything, but we like and report it because it complements the previous method.

Supp mat: "Even if reproductive isolation is complete, genome regions exhibiting greater population differentiation by chance (stronger drift) or due to divergent selection will produce greater cline width estimates." – perhaps I'm misunderstanding here but doesn't Supp Figure 6 show the opposite of this? That cline width measurements are lower for regions under selection or drift?

9) Thanks for spotting this, very much appreciated, we indeed got this wrong! We have corrected to *'lower cline width estimates'*.

I'm sure it was an oversight but there were no figure legends in the manuscript.

10) Sorry for that; it seems the legends were hidden due to some feature of the manuscript tracking system. We will ensure that the legends are present in this new submission.

Reviewer #3 (Remarks to the Author):

General comments

I consider that the revised manuscript “Clinal genomic analysis reveals strong reproductive isolation across a steep habitat transition in stickleback fish” by Haenel and co-authors has substantially improved when compared to the previous version. Namely, the implementation of a formal cline analysis contributed to strengthen the support for some of the authors’ claims. The clarifications of the study goals and the discussion on about the possible reproductive barriers on this system further contributed to increase the manuscript clarity and interest, respectively. However, I think that there are still some issues that need further attention.

First, the authors mention (several times) that the location of the main shift in allele frequencies and phenotypes coincide with the habitat transition. I touched on this issue in the previous round but I did not find the arguments provided by the authors very convincing (authors answer 57: “Well, we do not have spatially well-resolved phenotypic data, but given the available resolution, the inference that the phenotypic transition occur at the habitat transition seems uncontroversial to us. And there is common sense too; given the evidence of genome-wide RI at the habitat transition, why should the phenotypic transition be elsewhere?”). Please correct me if I am wrong, but is not the major habitat transition just downstream of S1 (i.e. between stream and marsh/lake)? If yes, the genome wide RI does not really coincide with the habitat transition. Or do the authors consider the main habitat transition between S1 and S2, even if these two are stream habitats? Furthermore, under the available resolution is actually difficult to objectively say where is the transition in term of number of gill rakers and pelvic spine length, but if I had to define visually a cline centre, I would perhaps put it at the left of the habitat transition (I may be wrong and the authors may have a different opinion but this just illustrates the difficulties of not being possible to use cline-fitting for the phenotypic data). Additionally, I think there are reasons why the phenotypic transition could differ from the habitat transition, such as directional dominance or reproductive isolation caused by intrinsic effects not related with those phenotypes (although this last hypothesis is perhaps less likely in this system based on the authors description), among others. Finally, the authors did not formally test if the cline centres differ across different classes of markers. Could not this be implemented? Based on all these arguments, I think the authors should at least be much more cautionary (and precise) when referring to coincidence among the main genetic, habitat and phenotypic transitions.

11) The first stream site S1 is *immediately* at the marsh-stream transition (merely a few meters upstream), so the habitat transition is indicated properly in our Fig. 2. Regarding the congruence between the habitat transition and the genetic cline, we emphasize that the genomic cline center is estimated to lie only some 30 m upstream of the habitat transition (see L165; visualized as red line in Fig .2). Sure, our study demonstrates that there is some gene flow over a small distance into the stream, thus shifting the genomic cline slightly upstream (for this reason, fish at S1 genetically resemble lake fish). But keep in mind that we are here dealing with a vertebrate capable of dispersing over a distance of 30 m within a few hours (L82-83). The spatial difference between the ecological and genomic transition is therefore trivial, and statements like our ‘*analyses reveal a remarkably tight association between ecology, phenotype, and genome-wide genetic variation*’ (L167-169) seem justified to us. Nevertheless, we agree with R3 that our interpretation that genomic transitions *coincide* with the habitat transition was not perfectly precise; they only *start* at the habitat transition. To clarify, we have improved

the text accordingly (L143, L159). Moreover, we have now added the specification that the stream site S1 is located right at the marsh-stream transition (L144-145).

Regarding R3's statement '*I think there are reasons why the phenotypic transition could differ from the habitat transition*' - the phenotypic data are too sparse for a precise localization, hence we see no reason to assume a difference in the location of these transitions. Also, we mention (L184-186) that there is *absolutely no* intrinsic reproductive isolation between these young ecotypes, so the possibility of '*intrinsic effects*' is not given. Moreover, all phenotypic differences between the ecotypes that have so far been scrutinized experimentally showed a strong genetic basis (we mention this very briefly on L66, with references). Overall, we feel there is good support that the phenotypic cline must broadly track the ecological and genomic one, and highlight that none of the three other evaluators questioned that both the genomic and phenotypic transitions are linked to the habitat shift. Nevertheless, we have now added a brief statement bringing to attention that around the habitat transition, the phenotypic data are sparse (L148-149), and we now use more cautious wording when inferring the similarity between the ecological and phenotypic transitions (L145-146).

Finally, we are surprised by R3's view that '*the authors did not formally test if the cline centres differ across different classes of markers*'. After all, our Fig. 4 presents a formal evaluation of cline center location based on compatibility intervals, following inferential strategies well rooted in the statistical literature (e.g., Manly 2006 *Randomization, bootstrap and Monte Carlo methods in biology*; Cumming et al. 2007 *J. Cell Biol*; Altman et al. 2013 *Statistics with Confidence: Confidence Intervals and Statistical Guidelines*; Halsey et al. 2019 *Biol. Lett.*).

Second, I am not an expert on cline analysis but I it is not clear for me if the authors first used cline-fitting to test if a given SNP has a clinal distribution or not. Actually, I am not sure if the package the authors used for cline-fitting allows to compare the likelihood between a clinal and non-clinal model (for instance using AIC)? However, this would allow to quantify the number or proportion of clinal SNPs across the three different classes of SNPs defined by the authors. Also, if a SNP is non clinal, I am not sure what the estimates of cline centers and widths mean. Furthermore, I found the suggestion of a cline-fitting artefact creating wider clines for median and loDiff SNPs supported by simulations an interesting possibility. This may well be the case. However, this is a bit challenging as that is also the expected pattern from residual gene flow (incomplete reproductive isolation). The authors fitted tailed clines, but I think this can be problematic for loci without fixed differences because it is hard to distinguish the tail from the end frequency. Thus, given the relatively small number of sites, would not make sense to use a model without tails to confirm that this is really an artefact?

12) No, we did not initially pre-select SNPs based on whether they formally qualify as clinal based on some threshold. We have checked the text and think that this is stated clearly in the methods (L535-537): '*For each SNP, cline fitting was run in ten replicates, and the median maximum likelihood estimate of cline center and cline width across these replicates was recorded*'. Moreover, the legend of Fig. 4 states that the '*median value across the 50 independent SNPs in each marker category*' is shown, again expressing that this analysis involved no exclusion of SNPs. We also do not believe that such a pre-selection would be adequate. The question addressed by our cline modeling is whether cline structure is similar between the loci with the strongest evidence of

selection versus the loci reflecting baseline drift across the entire genome. If for the latter, we first selected those SNPs qualifying most clearly as clinal, we would undermine this analysis because we would no longer be able to generalize estimated cline features to the entire genome. In short, this analysis is in our view performed in an unbiased and meaningful way appropriate to the study's objective. We also feel that in the previous round of review, R1 (Zach Gompert) had exactly this type of analysis in mind when suggesting that we should fit cline models '*for the fixed SNPs and a subset of random SNPs (e.g., the 500 random SNPs)*'.

As to R3's second concern, i.e., that shallower clines for SNPs exhibiting a relatively low allele frequency difference between the populations may not reflect an estimation artifact but ongoing gene flow, our simulations actually offer full clarification. As we emphasize in the legend of Supplementary Fig. 6, the three scenarios simulated (light gray, dark gray, and black) differ exclusively in the overall magnitude of population differentiation: '*the true cline position and width were thus tightly controlled and invariant among the different simulation types, which differed only in AFD between the populations.*' However, in a gene flow scenario as suggested by R3, the marker category showing the lowest magnitude of population differentiation should necessarily also show extended cline width, because reproductive isolation would be weaker for these genome regions. But again, we simulate identical cline widths among scenarios, and yet we observe an increase in estimated cline width with decreasing differentiation. In fact, as we declare in the last sentence of the legend to Supplementary Fig. 7, cline width estimates also become greater with decreasing differentiation in models with zero random noise; that is, even when allele frequencies display a perfect cline in each simulation group. The simulations thus demonstrate that the magnitude of differentiation alone – all else equal – drives cline width, and hence that differential cline width offers no convincing evidence of differential gene flow among our marker classes.

Finally, we do not share R3's concern about fitting tails because the empirical data show that allele frequencies stabilize rapidly with increasing distance from the cline center (see Fig. 2b for the selected and neutral SNPs, and Supplementary Fig. 4 for the loDiff SNPs). Nevertheless, we have now repeated both our empirical and simulation analyses based on models that do not assume tails, and we obtain results consistent with the tailed models; this model decision thus does not affect our conclusions in any relevant way. We now mention this new robustness check in the main Methods (L532-533) and in the legend of Supplementary Fig. 6. Finally, R3 feels that we use a relatively small number of sites. To address this concern, we have doubled the number of sites to a total of 20 in our simulations, and again obtain results fully consistent with those presented. This is now also mentioned in the legend of Supplementary Fig. 6.

The third issue is relative to one set of simulations implemented by the authors (L425-428). I brought this issue in the previous round (point vi) but I do not think I was clear enough. What I meant is that for the specific simulations the authors describe in L841-871, the authors basically let the populations evolve under drift and compare the empirical deltaAFD distribution with the distribution of values obtained with the simulations. However, I think that a model with drift alone is not the best null model. The authors mention in their answer (42) "If RI is genome-wide, the entire genome is in LD, hence EVERY base position is linked to selected loci (although of course not physically) ..." and in their answer (62) "Hence, no matter whether an individual dispersed only 150 m or 2 km from the lake into the stream (or vice versa), it would almost certainly be

eliminated by selection so rapidly that not even fragments of its genome managed to introgress.” Finally, the authors wrote in L.422-425: “This led us to hypothesize that our stream site closest to the lake (S1) was overwhelmed by gene flow from the lake, with appreciable genomic differentiation from the lake maintained in genome regions under the strongest divergent selection only. If true, these specific regions should, compared to randomly chosen genome regions, exhibit exceptionally strong differentiation between the lake and the stream population in general, including between the two samples from the endpoints of our geographic gradient (L1, S7).” Given this last statement and that there is genome-wide selection (two previous statements), I think that if the authors want to test whether those regions exhibit exceptionally strong differentiation compared with the rest of the genome, they should take into account that there is genome-wide selection when they let the populations evolve in their simulations. Any marker in the genome is under the influence of genome-wide selection and not just drift. Perhaps the conclusion does not change but I think that in this particular case, the influence of the genome-wide barrier should not be ignored.

13) Ok, thanks for pointing this out. We feel a key theoretical point underappreciated in evolutionary genomics is that selection primarily represents a reduction in effective population size, and reduced population size implies elevated drift. So if R3 suggests that we should not only consider genome-wide drift but also selection – this is exactly what our simulation model does; the genome-wide level of drift that we approximate includes the component driven by the population size reduction associated with the genome-wide selective barrier. The genome-wide effects of selection and drift therefore do not need to, and cannot, be disentangled. Our simulation approach is sound.

What was not optimal, however, was the wording chosen to describe the simulations. We have now improved by using an expression avoiding the artificial separation of selection from drift (L271-272).

Fourth, the authors mentioned in several answers that recombination is irrelevant in this particular study because reproductive isolation is complete (e.g. “Combined, this in our view offers compelling evidence that RI must be complete and homogeneous across the whole genome”, or essentially complete outside the narrow contact zone. However, either RI is complete or not. If it is incomplete in the contact zone. Then, it is incomplete and there is potential for alleles to recombine away from selected loci and introgress. Unfortunately, pool-seq data is not very informative about admixture. Even when we see clines of allele frequencies with pool-seq, we cannot distinguish, for instance, whether allele frequencies in a given point of the transect is 0.7 because all individuals have approximately 0.7 and 0.3 of ancestry from each source population, or because 70 individuals in the pool are homozygote for one allele and 30 individuals are homozygote for the other. Assuming that there is antagonism between selection and gene flow and that gene flow in contact zone is heterogeneous across the genome, as the authors defend, then I think that linked selection and recombination should be important, even if in a relatively narrow zone of the transect. The authors mention that there are genomic regions that resist gene flow (between L1 and S1). Thus, it would be important to discern if this is just due to selection or actually a combination of the both the effect of divergent selection and low recombination. I think the CCBD results somehow suggest that recombination plays a role in differentiation at least near the habitat transition. Probably this cannot be tested here but it is possible be that this antagonism between selection and effective migration (that also depends on recombination rate across the

genome) is important to avoid fusion around the habitat transition and that those highly differentiated regions can be instrumental to achieve a more genome-wide differentiation far from the center. Of course is just a hypothesis but I hope I presented valuable arguments why recombination may matter in this system.

14) We fully concur with R3 that it is somewhat awkward to use 'complete reproductive isolation' in a system in which a hybrid zone is demonstrated, even if this hybrid zone is incredibly narrow relative to the dispersal potential, and even if we clearly define what we mean by 'complete'. Given that R2 also found the qualifier 'complete' problematic, we have replaced this expression throughout the paper. We now use the more liberal expression 'strong reproductive isolation' (L52, L195, L218, L239).

Regarding the informativeness of poolSeq data about admixture, we think we already made an important distinction in our previous response letter: no matter if gene flow occurs exclusively in the form of direct immigration (not involving recombination) or in the form of true admixture (with hybridization and recombination), allele frequencies will always be affected. Hence gene flow along a cline is perfectly detectable with poolSeq data, but the degree of recombination is not. So if we observe that allele frequencies are completely stable beyond some distance from the habitat transition across all marker classes, this is solid evidence of a strong barrier to gene flow; beyond this point, recombination has no influence. We think that R3 agrees with us on this point.

So in this system, recombination can only have an interesting effect across the few hundred meters of hybrid zone. This is where we detect CCBD (R3 is correct that CCBD reflects an interaction between the recombination landscape and divergent selection; mentioned on L284-290). CCBD is the only inference on the role of recombination that our poolSeq data allow. Given that 1) recombination can only play a role over this short zone, that 2) our data do not allow digging deeper into details of recombination, and that 3) we added a statement during the last revision expressing that the breaking of haplotypes by recombination within the contact zone is an important avenue for future research (L307-311), we really think that all that needs to be said about recombination in this system is said. We do not see what could be added without turning speculative or recapitulating well-established theory.

Other comments

L100. I do not fully agree that variable progress across the speciation continuum necessarily supports primary divergence (L100). I am not saying that this system does not represent primary divergence but variable progress across the speciation continuum is also possible across multiple replicates of a secondary contact between two populations/species. This may happen if incompatibilities are polymorphic, where each replicate can follow its own evolutionary trajectory.

15) R2 raised exactly the same concern, and we agree. We have fixed this issue by giving a more convincing account of why lake-stream stickleback divergence must generally occur in the face of gene flow (i.e., primary divergence). See our response 2) above.

L273. Steep clines are a function of dispersal distance and do not necessarily translate into near complete reproductive isolation. Thus, inference of strong reproductive isolation should take into account dispersal. Is dispersal distance known for this system?

16) We explicitly cite empirical mark-recapture work documenting the dispersal of stickleback over hundreds of meters along streams within a few days (L81-83). That the

genomic cline requires strong reproductive isolation in this system is uncontroversial in our view. See also our response 1).

L346-348. Not all clines go to the bottom at S1, as expected given the patterns observed in Figure 3.

17) Apologies, but we do not understand to what text passage R3 here refers exactly; L346-348 in the pdf of the last manuscript version submitted deals with the inversions, which seems unrelated to the topic R2 raises here. But yes, there is a small proportion of genome regions that seem not fully overwhelmed by gene flow at site S1. The first part of the section titled '*Gene flow in the contact zone is heterogeneous across the genome*' discussed this pattern in detail (L246-278). We have checked the whole Results and Discussion part and see no inconsistency related to R3's comment.

L391-397. I do not follow why this asymmetry supports that there is reproductive isolation a few hundred meters above the habitat transition.

18) Again, this passage does not seem to be at the line numbers indicated by R3, but it is clear from the context what R3 refers to (the passage was located at L228-231 in the previous submission). The point here is not that asymmetry explains the precise location of the cline, but that asymmetry indicates that reproductive isolation must be *genome-wide*. We have made a small improvement to the text that should make this fully clear (L239).

L461-464. Is not also because there is actually low recombination and consequently less effective migration on those regions, even if foreign chromosome bits are not maladaptive?

19) As we understand, R3 here refers to a neutral scenario, that is, the absence of selection. But in the absence of selection, chromosome chunks of different lengths have similar survival probabilities, hence recombination has no consequences on allele frequencies and population differentiation. Extensive discussions and analyses are presented in Haenel et al. 2018 Mol. Ecol., and especially the theory paper by Berner & Roesti 2017 Mol. Ecol.. These references are cited (L286).

L510. Perhaps it would be useful to know the generation time of this species to know the time frame we are dealing here.

20) Good point, we have added (L63).

L537-580. I think it is good that the authors highlight the importance of polygenic selection in this system, as a contrast to the role of inversions in other systems, because it contributes to a more balanced view of the genomic architecture of reproductive isolation. However, the documented perturbation in the system could suggest that this polygenic differentiation is possibly reversible whereas this may be less likely with chromosomal rearrangements, where incompatibilities can accumulate despite of the gene flow in the rest of the genome (Navarro and Barton, 2003).

21) Yes, this sounds plausible. Indeed, the collapse of strongly reproductively isolated stickleback species pairs has been demonstrated (Taylor et al. 2006 Mol. Ecol.). However, we prefer not to address this topic in the text because we feel that to make a real contribution, one would need to formally compare models of divergence with polygenic variation versus models with rearrangement polymorphisms. And still, the

biological relevance of the latter is not fully clear to us - how could an organism achieve complex, multi-trait divergence (as occurs in stickleback) based on a rearrangement? We feel this issue goes beyond the scope of our study.

L679-682. Were no mapping quality filters applied? (for instance, to remove reads that map to multiple genomic regions)

22) Yes, sure, we only accepted unique alignment. In our early genomic studies, we used to mention this explicitly. But accepting reads with multiple alignment for genotyping makes no sense, hence the culture in the field seems to no longer state this explicitly (we confirmed by haphazardly checking a couple of recent genomics papers, including in Nat. Commun.). Also, we provide the full alignment options in the Supplementary Codes. This was not declared explicitly in the Methods, so we have added (L470-471). What is perhaps a bit less obvious is that even when non-unique alignments are excluded, a fraction of reads will still derive from repeats. This does not seem to be generally recognized, so we have expanded the text passage in which we describe how we filter such genome regions (L478-479).

L700. After reading the reason why the authors applied the extreme (0.25) MAF cut-off (answer 7), I wonder if this cannot contribute to bias towards a pattern of stronger whole-genome reproductive isolation. I know that a MAF of 0.25 does not mean that SNPs will be highly differentiated between habitats but could there be an effect of MAF on differentiation and clinal patterns mainly for the non-selected SNP set?

23) This is a question that comes up regularly within the context of MAF-filtering. Having devoted a methods paper specifically to this issue (Roesti et al. 2012 BMC Evol. Biol.), the last author of the present MS can guarantee that our MAF filter does not bias differentiation and clinal patterns. Instead, this strategy ensures that differentiation and clines are estimated with the most reliable, informative markers available. One should rather be worried if researchers did *not* filter by MAF; a striking and intuitive illustration of why MAF filtering is really needed when estimating differentiation is also provided in the Appendix S2 of Roesti et al. 2012 Mol. Ecol.. Given the above (well-cited) BMC methods paper, we prefer not to expand on this issue; we feel this would distract. However, this methods paper was not referred to in the previous MS, so we have added (ref 71, L480-481).

Other issues raised in the previous round that I would like to clarify (after '>')

ii) Another important aspect concerns stickleback migration. Do they migrate to breed in this lake/stream system? If they do, it would be important to know when was sampling conducted relative to breeding migration.

38. Migration: We declare that we are studying a 'population pair residing in parapatry (that is, contiguously) in Misty Lake and its inlet stream', which in our view is fairly explicit. Moreover, migration in this system is highly implausible biogeographically (Fig. 1a; where should the lake and stream populations go, and why?), so we prefer not to expand on this point further.

> Perhaps I was not clear but my question was if fish could migrate between lake and stream to breed in a different place from where they spend most of their life; if yes, perhaps it would be important to know when was sampling conducted relative to the time of the breeding migration.

24) Ok, good comment. There is no breeding migration; the fish breed where they live year-round, so in our view this does not warrant discussing in detail. Nevertheless, we agree that from a speciation perspective, it may be good to know that our study mirrors allele frequency patterns during the breeding season, as this is the life stage relevant to hybridization. So while we previously specified the sampling period only (L426), we have now added the information that this lies within the breeding period (L426-427).

iii) Finally, although I am fully aware of the problems related with FST, I am also not entirely sure that AFD (Berner, 2019) is free of biases. Would not be useful to compare the main results using AFD vs FST?

40. AFD vs FST: we are not clear what R3 means by 'biases' – we are interested in how allele frequencies change along a spatial gradient, hence AFD is the ideal differentiation metric directly expressing the quantity of interest. Nevertheless, we have now redone some key analyses based on FST, presented as Supplementary Figures 3 and 12.

> I meant that Berner (2019) also described some bias for AFD estimates, although it may not apply to this specific case, given the high sample sizes. Anyway, I think it is good that the authors now use both AFD and FST, so that the main results do not depend on the method used to estimate differentiation.

25) Fine, so this issue is closed.

iv) Given the effects of recombination (CCBD analysis), I would like to hear from the authors about the possibility of some of the AFD peaks resulting from background selection.

41) A discussion of background selection certainly has no place in this study...

> I thought could be beneficial to the reader that the authors just briefly explain why differentiations peaks they found are not likely generated by background selection but I am fine with the editor decision about this.

26) Background selection is really off topic here; any mutation-based process is irrelevant in a study system just a few thousand generations old. We hope R3 understands that we prefer not to invest space for discussing a mechanism unrelated to our results.

L60. Why only initiating?

48) Because even if ecologically-dependent RI is complete (= no more gene flow), speciation will typically remain reversible. Hence for complete speciation, sufficient intrinsic incompatibility is needed, which is a long-term process not necessarily requiring divergent selection.

> I think that a definition of speciation completeness that depends on irreversibility is not very useful because we will not be able to test it often. It also suggests that complete speciation is only possible if there are many intrinsic incompatibilities. However, this is just my opinion. I respect the authors' definition as soon as they are clear about it.

27) Appreciated; we agree that this specifier was not needed and was potentially confusing. We have clarified by deleting 'initiating' (L55-56).

L97. I think the authors should indicate here the number of samples used in each pool (average and range).

52) We say that N per site approximately 56 in the preceding sentence. Full details are given in Supplementary Table 1.

> I think I was not clear but I was wondering if average and range would not be more informative. Anyway, this is a minor detail as N is actually similar among sampling locations. Perhaps the authors should call Supplementary Table 1?

28) Good idea to refer to Supplementary Table 1 in this passage. We have added (L97).

L128. Not clear how the authors go from 34 to hundreds. Could not the target of selection be a single gene (or regulatory) per region?

... That hundreds of genome regions are under divergent selection in this system is uncontroversial; just eyeballing Fig. 3, one can discern some 20+ regions on chromosome I alone that must plausibly be under selection.

> I am not really saying that there are not hundreds of regions under selection. I actually do not know. However, I think there should be a more rigorous way of defining and quantifying them than just eyeballing.

29) Identifying regions under selection is a notorious challenge in genomics, and determining their precise number is impossible because regions under weak selection are lost in background differentiation. In our view, there simply is no rigorous way of defining such regions; one will always need to take decisions and apply thresholds, explicitly or tacitly. Nevertheless, we feel that from the evidence we present, readers will generally get a good intuitive sense for the polygenic nature of divergent selection in this system. We further emphasize that our wording is cautious in our view (L134), and that our conclusion agrees with strong evidence of highly polygenic divergence from other lake-stream stickleback systems (L135).

L482. Why were all loci unlinked?

66) Because these loci are simulation replicates, hence need to be independent.

> My question was not clear enough. I just wanted to ask why the authors did not modelled LD between markers, but it is now clear how this was done in the additional simulations with loci distributed across a single chromosome.

30) Fine, so this issue is closed.

Reviewers' Comments:

Reviewer #2:

Remarks to the Author:

This is my third time reading through the manuscript and I think the authors have done a good job of addressing my comments. I am glad they incorporated the ABC analysis to quantify the impact of the flooding - I think this is a nice touch.

I also see more clearly now why inversions are mentioned at the end of the MS. I agree, this is a useful balance.

A nice study, I enjoyed reviewing this one!

Reviewer #3:

Remarks to the Author:

Dear Daniel Berner and co-authors,

First, I would like to apologise for sending my report so late. This review caught me during a very busy period of two fieldwork trips (catching up from field trips that had been delayed due to the pandemic) and some illness issues, which limited a lot of my jobs during this period.

This is the third time I review the manuscript entitled "Clinal genomic analysis reveals strong reproductive isolation across a steep habitat transition in stickleback fish" by Haenel and co-authors (I was reviewer 3 in the last round). I think the authors did a good job in clarifying and addressing some important issues I raised before, inclusively by re-doing the cline fitting using different models, as well as being more cautionary and more precise in the description of some important patterns. Thus, I think that the manuscript (together with some authors' clarifications) has substantially improved to a point that it can be an interesting contribution to the field.

I have only two main doubts. First, I recognise that the authors have solid work on the effects of background selection/recombination on differentiation, that they are very knowledgeable on the study system, and I understand their arguments that this is a relatively young system. However, even if they are likely correct, I continue doubting if it would not be good to briefly explain in the text why background selection is unlikely to have generated at least some of the peaks. For example, could not some peaks result from sorting of older alleles, together with a reduction of within population diversity due to background selection? Or would this be a distraction (or off topic as the authors say)?

Secondly, it might have been more informative to consider heterogenous recombination rates to test the effects of recombination across the central part of the hybrid zone in their simulations. However, I understand that the authors consider that this deserves to be explored in more detail in the future using individual-based whole genome re-sequencing data.

It is not my intention to force the authors agree with me on these points and I have doubts myself whether this would have changed the overall conclusions (similar patterns are obtained with free recombination a single chromosome with crossover, as mentioned by the authors). I am just mentioning these two points so that the editor evaluates if it is essential to address them in a different manner before publication. Otherwise, I have only three minor additional comments:

1. Although the authors now use more cautious wording when they describe the similarity between, ecological, phenotypic and genomic transitions in the Introduction, I think the same should have been done in the abstract (L26: "co-localizes")
2. L81-84. Ok, but is this simply the potential for movement or is this a dispersal distance per generation?
3. L183-184 (habitat preference). It is not clear to me why habitat preference was neither included in the model nor its contribution to reproductive isolation discussed.

REVIEWERS' COMMENTS

Reviewer #2 (Remarks to the Author):

This is my third time reading through the manuscript and I think the authors have done a good job of addressing my comments. I am glad they incorporated the ABC analysis to quantify the impact of the flooding - I think this is a nice touch.

I also see more clearly now why inversions are mentioned at the end of the MS. I agree, this is a useful balance.

A nice study, I enjoyed reviewing this one!

We are happy that R2 appreciates our study, and we are grateful for the time this referee spent evaluating our work at different stages, and the thoughtful comments we received.

Reviewer #3 (Remarks to the Author):

Dear Daniel Berner and co-authors,

First, I would like to apologise for sending my report so late. This review caught me during a very busy period of two fieldwork trips (catching up from field trips that had been delayed due to the pandemic) and some illness issues, which limited a lot my jobs during this period.

This is the third time I review the manuscript entitled "Clinal genomic analysis reveals strong reproductive isolation across a steep habitat transition in stickleback fish" by Haenel and co-authors (I was reviewer 3 in the last round). I think the authors did a good job in clarifying and addressing some important issues I raised before, inclusively by re-doing the cline fitting using different models, as well as being more cautionary and more precise in the description of some important patterns. Thus, I think that the manuscript (together with some authors' clarifications) has substantially improved to a point that it can be an interesting contribution to the field.

I have only two main doubts. First, I recognise that the authors have solid work on the effects of background selection/recombination on differentiation, that they are very knowledgeable on the study system, and I understand their arguments that this is a relatively young system. However, even if they are likely correct, I continue doubting if it would not be good to briefly explain in the text why background selection is unlikely to have generated at least some of the peaks. For example, could not some peaks result from sorting of older alleles, together with a reduction of within population diversity due to background selection? Or would this be a distraction (or off topic as the authors say)?

As argued earlier, background selection is a very slow process based on deleterious *de novo* mutation, hence cannot be relevant to a study system just a few thousand generations old. Moreover, background selection cannot generate differentiation peaks, as it is a diffuse genome-wide process. Acknowledging R3's interest in background selection, we have decided not to address this topic in this final revision of our paper – we still feel that doing so would be distracting.

Secondly, it might have been more informative to consider heterogeneous recombination rates to test the effects of recombination across the central part of the hybrid zone in their simulations. However, I understand that the authors consider that this deserves to be explored in more detail in the future using individual-based whole genome re-sequencing data.

Yes, building on the present work, we indeed consider a follow-up study in which recombination rate will become a topic. But as we already mentioned, and as R3 recognizes (see his/her next passage), we checked for an effect of heterogeneous recombination rate in our simulations, and there is none. R3's intuition that this detail does not affect the conclusions of our study is undoubtedly correct. We see no need for further action here.

It is not my intention to force the authors agree with me on these points and I have doubts myself whether this would have changed the overall conclusions (similar patterns are obtained with free recombination a single chromosome with crossover, as mentioned by the authors). I am just mentioning these two points so that the editor evaluates if it is essential to address them in a different manner before publication.

Otherwise, I have only three minor additional comments:

1. Although the authors now use more cautious wording when they describe the similarity between, ecological, phenotypic and genomic transitions in the Introduction, I think the same should have been done in the abstract (L26: "co-localizes")

Ok, we see R3's point, although we feel that the original wording was unproblematic; we have rephrased the Abstract by using weaker wording (L26-27).

2. L81-84. Ok, but is this simply the potential for movement or is this a dispersal distance per generation?

We cannot see any ambiguity here, as this passage clearly mentions 'the potential of stickleback to disperse over hundreds of meters in a few days' (L82-83). We are confident that readers will generally understand that 'in a few days' is not 'per generation', especially since we mention that the typical life span of this animal is 1-2 years (L63).

3. L183-184 (habitat preference). It is not clear to me why habitat preference was neither included in the model nor its contribution to reproductive isolation discussed.

Regarding the inclusion of habitat preference in our model, a thorough theoretical analysis of habitat preference would be a topic big enough for a separate stand-alone paper, and clearly goes beyond the scope of the present study. Moreover, such a theoretical investigation has already been published – coincidentally by the last author of the present manuscript (Berner & Thibert-Plante 2015 *How mechanisms of habitat preference evolve and promote divergence with gene flow*. J. Evol. Biol.). Considering habitat preference in our theory part would therefore have added little novelty. Importantly, we do not know if habitat preference occurs in Misty lake-stream stickleback at all, and if it does, by what proximate mechanism it is caused. Expanding the treatment of habitat preference in our study would thus just add speculation and inflate the paper. We feel that our concise mention of habitat preference (L182-183), including a key reference, is adequate.